# ALIGNING LLM REASONERS BY META-LEARNING THE OPTIMAL IMITATION-EXPLORATION BALANCE

## ABSTRACT

Large Language Models (LLMs) are typically fine-tuned for reasoning tasks through a two-stage pipeline of Supervised Fine-Tuning (SFT) followed by Reinforcement Learning (RL), a process fraught with catastrophic forgetting and suboptimal trade-offs between imitation and exploration. Recent single-stage methods attempt to unify SFT and RL using heuristics, but lack a principled mechanism for dynamically balancing the two paradigms. In this paper, we reframe this challenge through the theoretical lens of implicit rewards, viewing SFT and RL not as distinct methods but as complementary reward signals. We introduce **Adaptive Meta Fine-Tuning (AMFT)**, a novel single-stage algorithm that learns the optimal balance between SFT's implicit, path-level reward and RL's explicit, outcome-based reward. The core of AMFT is a meta-gradient adaptive weight controller that treats the SFT-RL balance as a learnable parameter, dynamically optimizing it to maximize long-term task performance. This forward-looking approach, regularized by policy entropy for stability, autonomously discovers an effective training curriculum. We conduct a comprehensive evaluation on challenging benchmarks spanning mathematical reasoning, abstract visual reasoning (General Points), and vision-language navigation (V-IRL). AMFT consistently establishes a new state-of-the-art and demonstrats superior generalization on out-of-distribution (OOD) tasks. Ablation studies and training dynamic analysis confirm that the meta-learning controller is crucial for AMFT's stability, sample efficiency, and performance, offering a more principled and effective paradigm for LLM alignment. Our codes are open-sourced via `https://anonymous.4open.science/r/anonymous-amft-6B5B/`.

## 1 INTRODUCTION

Post-training fine-tuning is a critical stage for adapting Large Language Models (LLMs) to complex downstream tasks, such as multi-step reasoning and nuanced human preferences (Ouyang et al., 2022; Bai et al., 2022). The prevailing paradigm has been a sequential, two-stage pipeline: first, Supervised Fine-Tuning (SFT) on high-quality demonstrations, followed by Reinforcement Learning (RL) to optimize for a specific reward signal. While this SFT→RL pipeline has produced state-of-the-art models, it suffers from a fundamental tension. SFT excels at teaching models to imitate expert patterns but is confined to its static dataset, leading it to memorize rather than truly learn, thus failing to generalize to out-of-distribution scenarios (Chu et al., 2025; Rajani et al., 2025). Conversely, RL enables exploration beyond demonstrations, discovering novel solutions and generalizing better (Guo et al., 2025b). However, RL is notoriously sample-inefficient and unstable, especially under sparse rewards, and its on-policy nature limits its ability to learn beyond the base model's capabilities (Yan et al., 2025; Liu et al., 2025c). Most critically, the abrupt objective shift between stages often leads to catastrophic forgetting, where the RL stage overwrites the structured knowledge acquired during SFT (Chen et al., 2025; Chu et al., 2025).

This dilemma has spurred research into unified, single-stage frameworks that integrate SFT and RL more tightly. Recent approaches combine both objectives within a single training loop, using adaptive mechanisms to balance the two. These mechanisms, however, are fundamentally reactive, relying on short-term, heuristic signals. For instance, **SRFT** uses policy entropy (Fu et al., 2025), while **SuperRL**, **SASR**, and **DyME** employ dynamic switches based on reward density, gradient norms, or generation correctness, respectively (Liu et al., 2025c; Chen et al., 2025; Liu et al., 2025a). While these methods confirm the promise of hybrid training, they leave a crucial question unanswered:

how can we find the optimal balance between imitation and exploration in a principled and dynamic manner, rather than reacting to noisy, local signals?

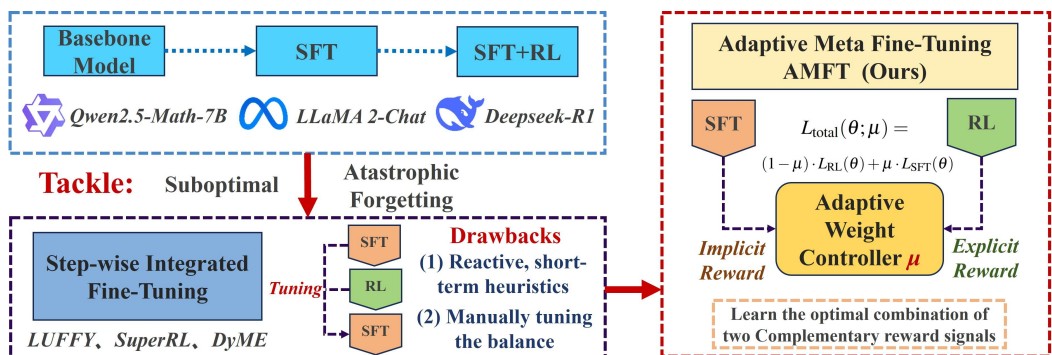

Figure 1: AMFT's motivation and framework. The method moves beyond the catastrophic forgetting of sequential pipelines and the reactive heuristics of prior single-stage methods by using a meta-controller to learn the optimal balance between SFT's implicit reward and RL's explicit reward.

In this paper, we reframe this challenge through the lens of **implicit reward learning**. Recent theoretical work has established that SFT is not merely distribution matching; it can be formally understood as a special case of RL that optimizes an implicit reward function encoded in expert demonstrations (Wang et al., 2025). This provides a powerful unified view: both SFT and RL are forms of reward optimization. SFT optimizes for an implicit, dense, path-level reward that encourages human-like reasoning structures, while RL optimizes for an explicit, often sparse, outcome-based reward that targets correctness. The challenge, therefore, is not to balance two different learning paradigms, but to learn the optimal combination of two complementary reward signals.

Building on this insight, we propose **Adaptive Meta Fine-Tuning (AMFT)**, a novel single-stage algorithm that addresses this challenge directly. The core of AMFT is an **adaptive weight controller** that treats the balance between the SFT and RL objectives as a learnable parameter, $\mu$. Unlike prior methods, our controller employs a meta-optimization strategy, updating $\mu$ using meta-gradients computed with respect to a long-term validation objective. This forward-looking approach effectively learns a dynamic training curriculum that maximizes final task performance. Intuitively, the controller learns to prioritize SFT's implicit reward when the policy is unstable, anchoring it to sound reasoning patterns. As the model gains competence, the controller shifts focus toward the explicit RL reward, encouraging exploration to discover higher-performing solutions.

Our contributions are as follows:

- We propose AMFT, a novel single-stage fine-tuning algorithm whose core innovation is a **meta-gradient-based adaptive weight controller**. This controller learns the optimal, dynamic balance between imitation (SFT) and exploration (RL) by directly optimizing for final task performance, moving beyond the reactive, heuristic-based mechanisms of prior work.

- We conduct a comprehensive evaluation on diverse benchmarks spanning **mathematical reasoning**, abstract visual reasoning (**General Points** (Zhai et al., 2024)), and vision-language navigation (**V-IRL** (Yang et al., 2024b)), demonstrating that AMFT consistently sets a new state-of-the-art.

- Our analysis of the training dynamics confirms that the meta-controller enables a more stable and efficient learning process, leading to superior generalization, particularly in multi-modal and sparse-reward settings.

## 2 RELATED WORK

Our work is situated at the intersection of supervised fine-tuning (SFT), reinforcement learning (RL), and meta-learning for LLM alignment. We contextualize our contributions by reviewing these three key areas.

Table 1: Systematic comparison of adaptive SFT-RL integration methods. Our proposed AMFT is the first to use a meta-gradient controller that directly optimizes for long-term performance, rather than relying on reactive, short-term heuristics.

| Method | Core Mechanism | Granularity | Limitation |
|---|---|---|---|
| SFT→RL (Baseline) | Sequential Training | Stage-level | Catastrophic forgetting; inefficient. |
| SRFT (Fu et al., 2025) | Single-Stage Weighted Sum | Continuous | Unreliable entropy heuristic. |
| SuperRL (Liu et al., 2025c) | Adaptive Switch | Step-level | Inflexible binary switch. |
| SASR (Chen et al., 2025) | Adaptive Integration | Step-level | Unreliable gradient norm proxy. |
| DyME (Liu et al., 2025a) | Dynamic Mode Selection | Step-level | Rigid switch; limited scope. |
| LUFFY (Yan et al., 2025) | Off-Policy Data Mixing | Step-level | Relies on fixed data ratio. |
| **AMFT (Ours)** | **Single-Stage Meta-Learning** | **Continuous** | **Principled, forward-looking optimization.** |

## 2.1 THE SFT-RL DICHOTOMY IN LLM ALIGNMENT

LLM post-training relies on two complementary paradigms. Supervised Fine-Tuning (SFT) aligns models by imitating expert demonstrations (Chung et al., 2022; Lewkowycz et al., 2022; Zhou et al., 2024), but its reliance on static data fosters memorization over generalization, degrading out-of-distribution (OOD) performance (Chu et al., 2025; Rajani et al., 2025). While theoretically viewed as optimizing an implicit reward, SFT provides a strong but insufficient initialization for robust reasoning (Wang et al., 2025). Conversely, Reinforcement Learning (RL) enables exploration beyond static data to optimize outcomes, with RL from Human Feedback (RLHF) aligning subjective preferences (Ouyang et al., 2022; Stiennon et al., 2020) and RL with Verifiable Rewards (RLVR) enhancing objective reasoning (Guo et al., 2025a; Team et al., 2025). However, RL is notoriously unstable and sample-inefficient; on-policy algorithms like PPO risk policy collapse (Schulman et al., 2017; Zhao et al., 2025; Yue et al., 2025; Ziegler et al., 2019), sparse rewards can cause advantage collapse (Liu et al., 2025a;c), and the process can lead to catastrophic forgetting of SFT-learned knowledge (Chen et al., 2025; Fernando et al., 2025). This tension between stable-but-brittle imitation and generalizable-but-unstable exploration motivates more integrated paradigms.

## 2.2 HYBRID PARADIGMS: THE PATH TO INTEGRATION

The drive to unify SFT and RL stems from their complementary strengths: SFT is highly efficient for distilling knowledge from static, offline expert data, whereas RL promotes generalization by exploring novel solutions through online interaction (Rajani et al., 2025; Chu et al., 2025). This has led to the development of hybrid paradigms, often termed mix-policy learning, which typically update the model using a composite objective function. Early strategies involved simple interleaving schedules or static loss combinations (Ma et al., 2025b; Liu et al., 2025b; Luong et al., 2024; Gulcehre et al., 2023). Recognizing that a fixed trade-off is suboptimal, a subsequent wave of adaptive frameworks emerged to dynamically adjust the balance. These methods, however, rely on various heuristic signals—such as policy entropy in SRFT (Fu et al., 2025), binary switches based on reward density in SuperRL and DyME (Liu et al., 2025c;a), or the gradient norm in SASR (Chen et al., 2025). While an improvement, these approaches are fundamentally reactive, adjusting the balance based on short-term proxy signals that may not correlate with final task performance (Yu et al., 2024; Kim et al., 2024). This leaves a crucial gap for a more principled, forward-looking strategy. To this end, our work introduces AMFT, which employs a meta-gradient controller (Franceschi et al., 2018) to autonomously learn an optimal, dynamic training curriculum, moving beyond reactive heuristics.

## 3 METHODOLOGY

We propose AMFT, a single-stage algorithm that unifies supervised fine-tuning and reinforcement learning. Our approach is grounded in a modern theoretical perspective that reframes the fine-tuning problem itself: rather than balancing two disparate learning paradigms, we see it as optimizing a single policy against two complementary reward signals.

### 3.1 A UNIFIED OBJECTIVE FOR IMITATION AND EXPLORATION

Our framework begins by formalizing SFT and RL under a common lens. We consider a policy $\pi_\theta$ that generates a trajectory $\tau$ (e.g., a reasoning chain) given an input $x$. This policy is guided by two distinct but valuable sources of information:

**Explicit Outcome-Based Reward (from RL).** The first signal is an explicit, verifiable reward $R_{\text{explicit}}(\tau)$, which evaluates the final outcome of a trajectory (e.g., $+1$ for a correct answer, $0$ otherwise). The standard RL objective is to maximize the expected value of this reward, encouraging the model to explore the solution space to find high-performing strategies:

$$J_{\text{RL}}(\theta) = \mathbb{E}_{\tau \sim \pi_\theta}[R_{\text{explicit}}(\tau)]. \tag{1}$$

In practice, we use a policy gradient loss, denoted $L_{\text{RL}}(\theta)$, such as the PPO-clip objective (Schulman et al., 2017), to perform gradient ascent on $J_{\text{RL}}(\theta)$.

**Implicit Path-Based Reward (from SFT).** The second signal is derived from a dataset of expert demonstrations, $\mathcal{D}_{\text{SFT}}$. Building upon the theoretical framework of implicit rewards (Wang et al., 2025; Cui et al., 2025a), we interpret the standard SFT loss (i.e., negative log-likelihood) not merely as imitation, but as the optimization of an implicit reward function, $R_{\text{implicit}}(\tau)$, which is high for trajectories that faithfully replicate the expert demonstrations. The SFT objective encourages the policy to align with these desirable paths:

$$L_{\text{SFT}}(\theta) = -\mathbb{E}_{(x, y_{\text{demo}}) \sim \mathcal{D}_{\text{SFT}}}[\log \pi_\theta(y_{\text{demo}}|x)]. \tag{2}$$

This objective promotes imitation, anchoring the policy to a distribution of human-aligned, structurally sound reasoning.

**The Unified Loss Function.** AMFT elegantly merges these two objectives into a single, dynamically weighted loss function:

$$L_{\text{total}}(\theta; \mu) = (1 - \mu) \cdot L_{\text{RL}}(\theta) + \mu \cdot L_{\text{SFT}}(\theta). \tag{3}$$

Here, the adaptive weight $\mu_t \in [0, 1]$ acts as a dynamic dial, controlling the relative influence of exploration versus imitation at each training step $t$. The central novelty of our work lies in how we learn the optimal schedule for $\mu_t$.

## 3.2 THE ADAPTIVE WEIGHT CONTROLLER: META-LEARNING THE OPTIMAL BALANCE

A fixed or manually scheduled $\mu$ is unlikely to be optimal. AMFT addresses this by introducing an **adaptive weight controller** that learns the optimal schedule for $\mu$ online, using a principled meta-optimization strategy. This elevates the balancing act to a bilevel optimization problem: the inner loop optimizes the policy parameters $\theta$ given a fixed $\mu$, while the outer loop optimizes $\mu$ to improve long-term performance.

**Long-Term Strategy via Meta-Gradient.** To ensure the learned schedule for $\mu$ is directly aligned with our ultimate goal, we treat $\mu$ as a learnable parameter and optimize it via a meta-gradient. We define a utility function $U(\theta)$ as the expected explicit reward on a held-out validation set $\mathcal{D}_{\text{val}}$:

$$U(\theta) = \mathbb{E}_{(x, \tau) \sim \pi_\theta(\cdot | \mathcal{D}_{\text{val}})}[R_{\text{explicit}}(\tau)]. \tag{4}$$

The controller periodically estimates the gradient of this utility with respect to $\mu$, effectively asking: "how will a small change in the current SFT/RL balance affect long-term validation performance?" This forward-looking signal is computed using the chain rule:

$$\nabla_\mu U(\theta_t) = \nabla_\theta U(\theta_t) \frac{\partial \theta_t}{\partial \mu}. \tag{5}$$

While computing the full Jacobian-vector product $\frac{\partial \theta_t}{\partial \mu}$ is expensive, we approximate it efficiently by differentiating a single step of the inner optimization update, $\theta_t \approx \theta_{t-1} - \alpha \nabla_\theta L_{\text{total}}(\theta_{t-1}; \mu_t)$, a common technique in meta-learning (Franceschi et al., 2018). This efficient one-step approximation nudges $\mu$ toward values expected to yield better future rewards.

**Synergizing Long-Term and Short-Term Control.** The meta-gradient provides a globally optimal direction for $\mu$ but is computationally expensive and estimated infrequently. It is therefore insufficient to handle immediate, step-level training instabilities. To address this, we supplement it with a fast-acting heuristic based on policy entropy, $H(\pi_\theta)$, which serves as a robust proxy for the model's uncertainty and stability.

---

**Algorithm 1** The AMFT Algorithm

---

**Require:** Pretrained model $\pi_\theta$; Demonstration data $\mathcal{D}_{\text{SFT}}$; Environment `env`; Initial weight $\mu_{\text{init}}$
**Ensure:** Fine-tuned model $\pi_\theta^*$
1: Initialize $\mu \leftarrow \mu_{\text{init}}$; Initialize value function $\phi$
2: **Warm-up Phase:** Train $\pi_\theta$ on $\mathcal{D}_{\text{SFT}}$ for $W$ steps using $L_{\text{SFT}}$.
3: **for** $t = 1$ **to** $T$ **do**
4:     Sample SFT batch $\{(x_i, y_i)\}_{i=1}^m \sim \mathcal{D}_{\text{SFT}}$ and compute $L_{\text{SFT}}$
5:     Sample RL rollouts $\{\tau_j\}_{j=1}^n \sim \pi_\theta$ in `env` and compute $L_{\text{RL}}$ using PPO objective
6:     *// Update adaptive weight $\mu_t$*
7:     Compute meta-gradient $g_\mu \leftarrow \nabla_\mu U(\theta_t)$ on a validation batch (periodically)
8:     Compute entropy heuristic $g_H \leftarrow H^* - H(\pi_{\theta_t})$
9:     $\mu_{t+1} \leftarrow \text{clip}(\mu_t + \eta_\mu g_\mu + \eta_H g_H, \mu_{\min}, \mu_{\max})$
10:    *// Update model parameters*
11:    $L_{\text{total}} \leftarrow (1 - \mu_{t+1}) \cdot L_{\text{RL}} + \mu_{t+1} \cdot L_{\text{SFT}}$
12:    Update policy $\theta_{t+1}$ and value function $\phi_{t+1}$ by descending $\nabla L_{\text{total}}$ and $\nabla L_{\text{value}}$
13: **end for**
14: **return** $\pi_\theta$

---

- **High Entropy** ($H(\pi_\theta) \gg H^*$)**:** Indicates policy uncertainty or chaotic exploration. The controller increases $\mu$ to strengthen the stabilizing influence of the SFT loss.

- **Low Entropy** ($H(\pi_\theta) \ll H^*$)**:** Suggests the policy is becoming too deterministic, risking overfitting or policy collapse. The controller decreases $\mu$ to encourage exploration.

The target entropy $H^*$ is a hyperparameter, initialized based on the average entropy of the warm-up SFT policy. The final update rule for $\mu$ synergistically combines these two signals:

$$\mu_{t+1} = \text{clip}\left(\mu_t + \eta_\mu \nabla_\mu U(\theta_t) + \eta_H(H^* - H(\pi_{\theta_t})), \mu_{\min}, \mu_{\max}\right), \quad (6)$$

where $\eta_\mu$ and $\eta_H$ are learning rates for the long-term meta-gradient and short-term entropy heuristic, respectively. This dual-mechanism controller allows AMFT to pursue a long-term optimal strategy while deftly navigating short-term instabilities.

## 3.3 THE AMFT ALGORITHM IN PRACTICE

Algorithm 1 outlines the complete single-stage training loop. The process begins with a brief SFT **warm-up** phase, which provides a stable and instruction-aligned initialization. In the main loop, each update step uses a mixed batch of data from both the SFT dataset and on-policy rollouts. The controller first updates the weight $\mu_t$. This new weight is then used to compute the unified loss $L_{\text{total}}$, which in turn updates the model's parameters. This tight, single-loop integration ensures that every gradient step is informed by an up-to-date assessment of the optimal balance. More implementation details are in AppendixC.

## 3.4 THEORETICAL GROUNDING

Our adaptive framework is grounded in established theoretical principles. The SFT loss term, $\mu \cdot L_{\text{SFT}}$, can be interpreted as a proxy for a dynamic Kullback-Leibler (KL) divergence penalty, $D_{\text{KL}}(\pi_{\text{demo}} \| \pi_\theta)$, that regularizes the policy $\pi_\theta$ against deviating too far from the expert demonstration distribution $\pi_{\text{demo}}$. We provide an intuitive sketch of this connection here and defer the full mathematical derivations to AppendixB.

The SFT objective minimizes the negative log-likelihood: $\min_\theta -\mathbb{E}_{\tau \sim \pi_{\text{demo}}}[\log \pi_\theta(\tau|x)]$. This is equivalent to maximizing the log-probability of the demonstration data. Given that the entropy term $\mathbb{E}_{\tau \sim \pi_{\text{demo}}}[\log \pi_{\text{demo}}(\tau|x)]$ is constant with respect to the model parameters $\theta$, this maximization is mathematically equivalent to minimizing the KL divergence $D_{\text{KL}}(\pi_{\text{demo}} \| \pi_\theta)$. Thus, the SFT loss acts as a data-driven KL regularizer. In this view, our adaptive controller is effectively learning the optimal, time-varying Lagrange multiplier ($\mu$) for this KL constraint, making it more principled and responsive than the fixed KL penalty commonly used in RLHF (Ziegler et al., 2019).

## 4 EXPERIMENTS

### 4.1 EXPERIMENTAL SETUP

**Training Datasets and Implementation Details.** We fine-tune **Qwen2.5-Math-7B** (Yang et al., 2024a) for mathematical reasoning and **LLaMA-3.2-Vision-11B** (AI@Meta, 2024) for visual tasks. For math, we use the OpenR1-Math-46k-8192 [1] (Yan et al., 2025) dataset, leveraging its prompts for RL rollouts and its high-quality solutions for SFT-based objectives. For visual reasoning, we use the official training splits from the **General Points** (Zhai et al., 2024) and **V-IRL** (Yang et al., 2024b) benchmarks. To ensure a fair comparison of convergence, all RL-based methods are trained for 500 steps using the PPO algorithm with 8 rollouts per prompt. Further details are in the AppendixD.

**Evaluation Benchmarks and Metrics.** Our evaluation is extensive and multi-faceted. For mathematical reasoning, we report in-distribution (ID) performance on five benchmarks: AIME24 (Li et al., 2024), AMC (Li et al., 2024), MATH500 (Hendrycks et al., 2021), Minerva (Lewkowycz et al., 2022), and OlympiadBench (He et al., 2024). Generalization is measured on three out-of-distribution (OOD) benchmarks: ARC-C (Clark et al., 2018), GPQA-D (Rein et al., 2024), and MMLU-Pro (Wang et al., 2024). For visual reasoning (General Points and V-IRL), we evaluate both ID and OOD performance, where OOD variants test generalization to novel rules or visual features. During inference, we set the generation temperature to 0.6 and a maximum sequence length of 8,192 tokens. We employ Math-Verify for reward computation during training and the OAT-Grader (Liu et al., 2024) for final evaluation.

**Baseline Methods.** We benchmark AMFT against a comprehensive suite of baselines using Qwen2.5-Math-7B.

- **Standard Paradigms:** We include **SFT-only** on the training data; **RL-only** (GRPO) trained from scratch; and the sequential **SFT→RL** pipeline.

- **State-of-the-Art Hybrid Methods:** We compare against leading single-stage frameworks: **LUFFY** (Yan et al., 2025), a mixed-policy GRPO approach using off-policy data; **ReLIFT** (Ma et al., 2025a), which interleaves RL with online fine-tuning on hard questions; **TAPO** (Wu et al., 2025), which integrates external knowledge as thought patterns into GRPO; and **SRFT** (Fu et al., 2025), which unifies SFT and RL through entropy-aware weighting.

### 4.2 RESULTS ON MATHEMATICAL REASONING

**Quantitative Performance.** As presented in Table 2, AMFT consistently establishes a new state-of-the-art. It achieves the highest average accuracy on both the five in-distribution (ID) math benchmarks (**61.3%**) and the three out-of-distribution (OOD) general reasoning benchmarks (**63.3%**). This demonstrates superior overall performance and generalization. This balanced excellence is directly attributable to the meta-gradient controller, which learns to retain sufficient SFT guidance to preserve general knowledge (preventing catastrophic forgetting on OOD tasks) while still aggressively optimizing in-domain reasoning performance through exploration. reasoning performance via exploration.

**Analysis of Training Dynamics.** The learning trajectories in Figure 2 and Figure 3 reveal why AMFT succeeds. Sequential SFT→RL methods are highly sensitive to the SFT duration, a manually-tuned hyperparameter that is difficult to optimize (Figure 2). In contrast, AMFT's adaptive weight $\mu$ (red dash-dotted line) autonomously learns the optimal curriculum, smoothly transitioning from an SFT-dominant to an RL-dominant phase. Furthermore, Figure 3 shows that while pure RL quickly suffers from policy collapse (indicated by rapidly decreasing entropy and stagnating rewards), AMFT's controller injects stabilizing SFT guidance to maintain high entropy. This sustained exploration allows AMFT to operate in a high-reward, high-entropy space, leading to more robust and higher-performing final policies. Please refer to Appendix F for further studies on the AMFT Controller.

---

[1] `https://huggingface.co/datasets/Elliott/Openr1-Math-46k-8192`

Table 2: Results on mathematical reasoning (in-distribution) and general reasoning (out-of-distribution) benchmarks. All models are trained from Qwen2.5-Math-7B. Accuracy (%) is reported. '*' indicates results are taken from the corresponding paper.

| Model | In-Distribution Performance | | | | | | Out-of-Distribution Performance | | | |
|---|---|---|---|---|---|---|---|---|---|---|
| | AIME24 | AMC | MATH500 | Minerva | Olympiad | Avg. | ARC-C | GPQA-D | MMLU-Pro | Avg. |
| Qwen2.5-Math | 11.5 | 31.6 | 46.7 | 7.9 | 15.8 | 22.7 | 18.0 | 11.1 | 16.7 | 15.2 |
| Qwen2.5-Math-Instruct | 12.4 | 48.3 | 80.4 | 33.0 | 39.2 | 42.6 | 70.1 | 24.2 | 34.1 | 42.8 |
| *Supervised Fine-Tuning* | | | | | | | | | | |
| SFT | 31.0 | 62.4 | 84.9 | 39.0 | 53.1 | 54.1 | 76.1 | 25.7 | 45.1 | 49.0 |
| SFT$_{KL}$ | 12.9 | 45.1 | 69.8 | 26.4 | 36.0 | 38.0 | 33.1 | 22.2 | 30.3 | 28.5 |
| *Reinforcement Learning* | | | | | | | | | | |
| RL$_{GRPO}$ (Shao et al., 2024) | 24.1 | 61.5 | 79.0 | 32.9 | 47.1 | 48.9 | 75.2 | 30.9 | 41.7 | 49.3 |
| SimpleRL-Zero* (Zeng et al., 2025) | 27.0 | 54.9 | 76.0 | 25.0 | 34.7 | 43.5 | 30.2 | 23.2 | 34.5 | 29.3 |
| PRIME-Zero* (Cui et al., 2025b) | 17.0 | 54.0 | 81.4 | 39.0 | 40.3 | 46.3 | 73.3 | 18.2 | 32.7 | 41.4 |
| OpenReasoner-Zero* (Hu et al., 2025) | 16.5 | 52.1 | 82.4 | 33.1 | 47.1 | 46.2 | 66.2 | 29.8 | **58.7** | 51.6 |
| *SFT and RL* | | | | | | | | | | |
| SFT → RL | 32.0 | 66.9 | 84.1 | 34.1 | 56.2 | 54.6 | 76.3 | 37.8 | 49.6 | 54.6 |
| LUFFY (Yan et al., 2025) | 29.4 | 65.5 | 87.2 | 37.3 | 57.2 | 55.3 | 80.1 | 39.5 | 52.6 | 57.4 |
| TAPO* (Wu et al., 2025) | 33.3 | 77.5 | 83.4 | 38.2 | 46.2 | 55.7 | 81.6 | 37.9 | 49.6 | 56.4 |
| ReLIFT* (Ma et al., 2025a) | 28.2 | 64.8 | 85.0 | 37.1 | 54.9 | 54.0 | 74.9 | 40.9 | 51.9 | 55.9 |
| SRFT* (Fu et al., 2025) | 35.3 | 74.3 | **89.8** | 39.7 | 58.3 | 59.5 | **85.3** | 46.4 | 55.9 | 62.5 |
| **AMFT (ours)** | **36.1** | **77.9** | 89.5 | **40.9** | **62.1** | **61.3** | 84.1 | **47.5** | 58.3 | **63.3** |

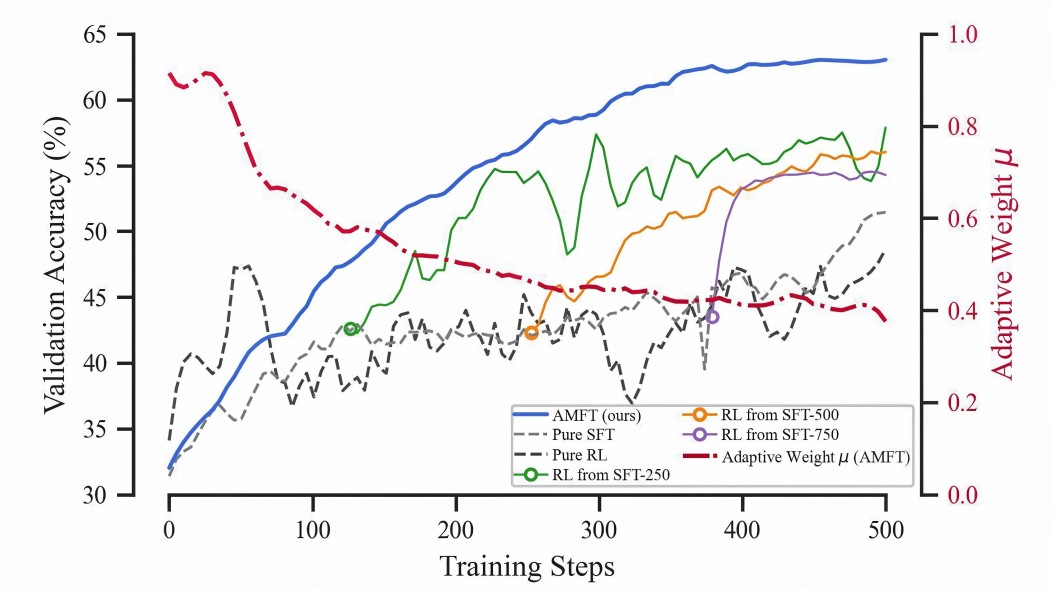

Figure 2: AMFT learning dynamics vs. sequential baselines on math benchmarks. The left y-axis shows validation accuracy (%), while the right y-axis shows the adaptive weight $\mu$. AMFT (solid blue) achieves a superior learning curve by dynamically adjusting $\mu$ (red dash-dotted), avoiding the difficult and manually-tuned trade-offs of sequential SFT→RL methods.

### 4.3 RESULTS ON VISUAL REASONING AND GENERALIZATION

To validate AMFT in multi-modal settings, we test its ability to resolve the *SFT Memorizes, RL Generalizes* dilemma (Chu et al., 2025).

**In-Distribution vs. Out-of-Distribution Performance.** We evaluated each method on ID and OOD versions of the General Points and V-IRL tasks. As presented in Table 3, the results are conclusive. The baseline methods clearly illustrate the fundamental trade-off: **SFT-only** performs reasonably in-distribution but its performance collapses on OOD tasks. Conversely, **RL-only** shows much stronger OOD performance but lags behind SFT on ID tasks. The two-stage **RL-from-SFT** and off-policy **LUFFY** offer progressively better compromises.

However, **AMFT consistently achieves the best performance across all conditions**. It not only sets the highest score on ID tasks but also exhibits the most robust OOD generalization, with the smallest

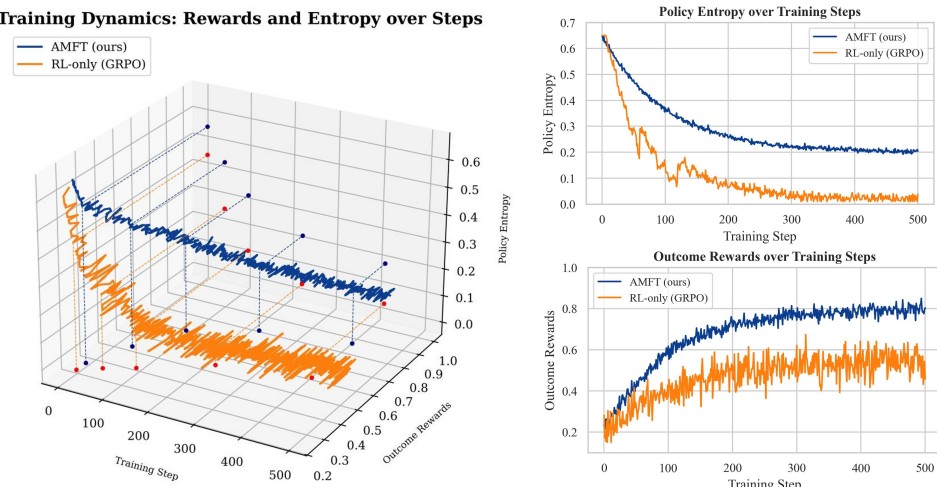

Figure 3: Comparative analysis of training dynamics between AMFT and a pure RL-only (GRPO) baseline on mathematical reasoning tasks. The main 3D visualization (left) plots the learning trajectories across training steps, outcome rewards, and policy entropy. For clarity, 2D projections for policy entropy (top right) and outcome rewards (bottom right) are provided. The plots reveal two distinct behaviors: the RL-only policy rapidly converges to a low-entropy state (policy collapse), limiting its reward potential.

Table 3: In-distribution (ID) and Out-of-distribution (OOD) performance on visual reasoning tasks. Win/Success rates (%) are reported for both rule and visual generalization. The data reflects the principle that SFT excels in-distribution while RL generalizes better out-of-distribution, with AMFT achieving the best of both.

| Method | General Points (Visual) | | | V-IRL Navigation | | |
|---|---|---|---|---|---|---|
| | ID Win% | OOD (Rule)% | OOD (Visual)% | ID Success% | OOD (Rule)% | OOD (Visual)% |
| SFT-only | 22.5 | 5.6 | 13.7 | 88.0 | 2.5 | 11.1 |
| RL-only (from scratch) | 41.2 | 14.2 | 41.2 | 85.0 | 45.0 | 65.0 |
| RL-from-SFT (two-stage) | 55.0 | 25.5 | 52.0 | 92.5 | 55.1 | 77.8 |
| LUFFY (Yan et al., 2025) | 62.3 | 35.2 | 61.5 | 94.0 | 64.8 | 82.1 |
| **AMFT (ours)** | **72.1** | **45.8** | **70.3** | **95.2** | **71.4** | **85.2** |

Table 4: Ablation study results across all task domains. Performance is reported as the primary metric for each task (%). All components of the AMFT controller are shown to be critical.

| AMFT Variant | Math Reasoning Acc. (%) | | General Points (Visual) | | | V-IRL Navigation | | |
|---|---|---|---|---|---|---|---|---|
| | ID Avg. | OOD Avg. | ID Win% | OOD (Rule)% | OOD (Visual)% | ID Success% | OOD (Rule)% | OOD (Visual)% |
| **AMFT (Full)** | **61.3** | **63.3** | **72.1** | **45.8** | **70.3** | **95.2** | **71.4** | **85.2** |
| w/o Meta-Gradient (Entropy-only) | 57.0 | 60.5 | 68.1 | 41.5 | 66.2 | 92.3 | 67.8 | 81.0 |
| w/o Entropy Heuristic (Meta-only) | 55.0 | 57.5 | 65.4 | 38.2 | 62.1 | 88.5 | 62.1 | 75.3 |
| w/o SFT Warm-up | 56.2 | 58.8 | 62.3 | 35.6 | 58.5 | 85.1 | 60.5 | 72.4 |

relative performance drop. This demonstrates that AMFT's adaptive controller learns to leverage SFT's structural guidance to build a strong in-distribution foundation while seamlessly transitioning to RL-driven exploration to learn the underlying task logic required for OOD success. It does not just combine SFT and RL; it learns the optimal curriculum for integrating them, thereby achieving the best of both worlds.

**Ablation Study and Efficiency Analysis.** To dissect AMFT's core components, our ablation study (Table 4) confirms that all parts are essential and synergistic. Removing the meta-gradient ('w/o Meta-Gradient') causes a consistent performance drop, underscoring that the forward-looking, performance-driven signal is crucial for discovering an optimal curriculum. Meanwhile, removing the entropy heuristic ('w/o Entropy Heuristic') leads to even greater degradation and training instability, confirming its necessity as a reactive regularizer for short-term stability. Furthermore, eliminating the initial SFT warm-up ('w/o SFT Warm-up') significantly harms performance, proving that a stable,

Table 5: Computational and sample efficiency analysis on **OOD (Visual)** benchmarks. We report the resources required to reach a target performance (60% win rate on GP-Visual, 70% success rate on V-IRL). AMFT demonstrates significant gains in both computational (fewer steps) and sample efficiency (fewer expensive RL rollouts). Peak performance is reported for methods that failed to reach the target.

| Method | General Points (Visual) — Target: 60% Win Rate | | | | V-IRL Navigation — Target: 70% Success Rate | | | |
| --- | --- | --- | --- | --- | --- | --- | --- | --- |
| | # Training Steps | # SFT Samples | # RL Rollouts | Peak Perf. (%) | # Training Steps | # SFT Samples | # RL Rollouts | Peak Perf. (%) |
| *Baselines* | | | | | | | | |
| SFT-only | — | ~150,000 | 0 | 13.7 | — | ~120,000 | 0 | 11.1 |
| RL-from-scratch | ~480 | 0 | ~30,720 | 60.0 | — | 0 | >90,000 | 48.2 (DNC*) |
| *Two-Stage & Hybrid Methods* | | | | | | | | |
| RL-from-SFT | ~420 | 60,000 (fixed) | ~21,760 | 60.0 | — | 50,000 (fixed) | >75,000 | 68.5 (DNC*) |
| LUFFY (Yan et al., 2025) | ~400 | ~32,000 | ~22,400 | 60.0 | ~450 | ~36,000 | ~44,800 | 70.0 |
| **AMFT (ours)** | **~310** | **~49,600** | **~15,840** | **60.0** | **~340** | **~54,400** | **~30,720** | **70.0** |

*DNC: Did Not Converge. Performance plateaued without reaching the target.

instruction-aligned starting point is vital for effective exploration, particularly in complex visual domains.

**Computational Cost and Sample Efficiency.** To quantify AMFT's efficiency, we measured the resources required to reach demanding performance thresholds on the challenging **Out-of-Distribution (OOD) Visual variants** of our multi-modal benchmarks: a 60% win rate on General Points and a 70% success rate on V-IRL Navigation. We assess efficiency via computational cost (# Training Steps) and sample efficiency, distinguishing between inexpensive SFT samples and costly RL rollouts. The results in Table 5 confirm AMFT's superiority. While baselines like SFT-only and RL-from-scratch fail to reach these OOD targets, AMFT converges with the fewest training steps. Most critically, it dramatically reduces the number of expensive RL rollouts by intelligently substituting them with cheaper SFT updates via its adaptive controller. This principled resource management confirms that AMFT provides a more practical and scalable path to robust generalization. Please refer to the appendix E for more details on the experimental results and analysis.

# 5 CONCLUSION

In this work, we introduced Adaptive Meta Fine-Tuning (AMFT), a single-stage algorithm that moves beyond the reactive heuristics of prior methods. By reframing SFT and RL as the optimization of complementary implicit and explicit reward signals, AMFT utilizes a forward-looking meta-gradient controller to learn the optimal, dynamic balance between them. This approach autonomously discovers an effective training curriculum, demonstrably preventing catastrophic forgetting and policy collapse. Our comprehensive evaluations show that AMFT establishes a new state-of-the-art in performance, generalization, and sample efficiency across diverse mathematical and visual reasoning benchmarks, paving a more principled path toward autonomous model alignment.

# 6 DISCUSSION

AMFT's forward-looking optimization contributes to the central goal of enhancing LLM reasoning (Jaech et al., 2024), but it also illuminates critical trade-offs and future challenges for the field. The meta-gradient's slightly computational overhead presents a potential scalability challenge for larger model, highlighting a core tension between principled, theoretically-grounded alignment and the practical, resource-efficient pipelines required for frontier models (Yoshihara et al., 2024). Beyond mere efficiency, AMFT's reframing of SFT as an implicit reward enriches the discourse on unifying imitation and exploration (Zeng et al., 2024), but it also surfaces a crucial concern: the learned curriculum is critically dependent on the quality and representativeness of the validation data. This vulnerability means that a misspecified or biased validation set could lead the model to perfectly learn a curriculum for a flawed objective, a subtle but dangerous form of specification gaming that current safety paradigms struggle to address. This positions the algorithm as a step toward more autonomous systems that learn how to learn, a concept being actively explored through mechanisms like policy-based self-correction (Zhang et al., 2024). Ultimately, the path forward requires not only balancing principled optimization with scalability but also developing meta-learning frameworks that are robust to objective misspecification. The true frontier is not just building models that can reason, but architecting systems that can autonomously and safely refine their own learning processes, navigating the complex landscape of human values.

## ETHICS STATEMENT

The work presented in this paper is methodological in nature, focusing on the development of algorithms for large language models. To the best of our knowledge, our proposed methods do not introduce any new ethical concerns.

## REPRODUCIBILITY STATEMENT

To facilitate the verification of our results, the implementation code for our algorithm and the main baselines is provided in the anonymous code link and the appendix.

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

APPENDIX

## A   USE OF LARGE LANGUAGE MODELS

We utilized a large language model to enhance the language and clarity of our manuscript. Specifically, we employed Gemini 2.5 flash with the following prompt to refine the initial draft: *I am writing an academic paper in English. Please polish the following draft so that it adheres to the conventions of academic writing.*

## B   THEORETICAL FOUNDATIONS OF AMFT

This section provides the rigorous mathematical derivations that underpin AMFT's core premise: reframing SFT and RL as the optimization of complementary reward signals. We build upon the theoretical framework established by (Wang et al., 2025) to formalize the connection between imitation learning (SFT) and implicit reward optimization.

### B.1   FULL DERIVATION OF SFT AS IMPLICIT REWARD OPTIMIZATION

The goal of this section is to formally demonstrate that the standard Supervised Fine-Tuning (SFT) objective is a special case of a broader reinforcement learning framework aimed at optimizing an implicit reward function (Wang et al., 2025; Cui et al., 2025a). This unified perspective is central to the design of AMFT.

Our derivation begins with a general objective for imitation learning: minimizing the divergence between the state-action distribution of a policy $\pi$ ($\mu_\pi$) and that of an expert ($\mu_E$). In the context of LLM post-training, it is crucial not to deviate excessively from the knowledge and linguistic priors of the base model, $\pi_{\text{ref}}$. Therefore, we modify the standard entropy regularizer to a Kullback-Leibler (KL) divergence term against the reference policy (Wang et al., 2025), yielding the objective:

$$\min_\pi D_f(\mu_\pi || \mu_E) + \beta D_{KL}(\pi || \pi_{\text{ref}}) \tag{7}$$

where $D_f$ is a chosen f-divergence and $\beta$ is the regularization coefficient.

Following the principles of non-adversarial imitation learning (Wulfmeier et al., 2024; Garg et al., 2022), this objective can be transformed into an equivalent min-max problem using the convex conjugate function, $f^*$. This allows us to re-express the divergence minimization problem as a reward optimization problem:

$$\min_\pi D_f(\mu_\pi || \mu_E) + \beta D_{KL}(\pi || \pi_{\text{ref}})$$
$$= \min_\pi \max_g \{\mathbb{E}_{\mu_\pi}[g] - \mathbb{E}_{\mu_E}[f^*(g)]\} + \beta D_{KL}(\pi || \pi_{\text{ref}}) \tag{8}$$

where $g : \mathcal{S} \times \mathcal{A} \to \mathbb{R}$ is an arbitrary function. By substituting $g = -r$ to align with the standard RL reward maximization convention and leveraging the saddle-point property of the objective, we can swap the min and max operators (Wang et al., 2025):

$$-\min_r \left[ \mathbb{E}_{\mu_E}[f^*(-r)] + \max_\pi (\mathbb{E}_{\mu_\pi}[r] - \beta D_{KL}(\pi || \pi_{\text{ref}})) \right] \tag{9}$$

The inner maximization problem, $\max_\pi (\mathbb{E}_{\mu_\pi}[r] - \beta D_{KL}(\pi || \pi_{\text{ref}}))$, is a standard regularized RL objective. As demonstrated in prior work (Rafailov et al., 2024), this problem has a closed-form solution where the value of the objective function at the optimal policy $\pi^*$ is the optimal value function at the initial state, $V^*(s_0)$.

Furthermore, the relationship between the optimal policy $\pi^*$ and the corresponding reward function $r$ is given by (Rafailov et al., 2024):

$$r(x, y) = \beta \log \frac{\pi^*(y|x)}{\pi_{\text{ref}}(y|x)} + V^*(s_0) - V^*(s_t) \tag{10}$$

where $s_t$ denotes a state in the trajectory $y$.

To complete the derivation and connect this general framework to SFT, we select a specific f-divergence. Following (Wang et al., 2025), we choose the Total Variation (TV) distance. The convex conjugate for the TV distance is the identity function, $f^*(t) = t$. Substituting this into Eq. 9 and focusing on the inner maximization over $\pi$, the objective becomes:

$$\max_{\pi,r} [\mathbb{E}_{\mu_E}[-f^*(-r)] - V_\pi(s_0)]$$
$$= \max_{\pi,r} [\mathbb{E}_{\mu_E}[r] - V_\pi(s_0)] \tag{11}$$

Now, we substitute the reward $r$ in Eq. 11 with its policy-dependent form from Eq. 10. This yields:

$$\max_\pi \mathbb{E}_{\mu_E} \left[ \beta \log \frac{\pi(y|x)}{\pi_{\text{ref}}(y|x)} + V_\pi(s_0) - V_\pi(s_t) \right] - V_\pi(s_0)$$
$$= \max_\pi \mathbb{E}_{(x,y)\sim\mathcal{D}_{\text{SFT}}} \left[ \beta \log \frac{\pi(y|x)}{\pi_{\text{ref}}(y|x)} - V_\pi(s_t) \right] \tag{12}$$

Since $\pi_{\text{ref}}$ and $V_\pi(s_t)$ are constant with respect to the optimization of $\pi(y|x)$ at each step, maximizing this objective is equivalent to maximizing $\mathbb{E}_{(x,y)\sim\mathcal{D}_{\text{SFT}}}[\log \pi(y|x)]$. This is precisely the objective of minimizing the standard SFT loss (negative log-likelihood):

$$L_{\text{SFT}}(\theta) = -\mathbb{E}_{(x,y)\sim\mathcal{D}_{\text{SFT}}}[\log \pi_\theta(y|x)] \tag{13}$$

This derivation formally establishes that SFT is a special case of implicit reward optimization, where the implicit reward function being learned is identical in form to that of preference learning methods like DPO (Wang et al., 2025). This provides the theoretical basis for our unified view of SFT and RL as optimizing complementary reward signals within a single framework.

### B.2  THEORETICAL JUSTIFICATION OF SFT LOSS AS A KL DIVERGENCE PROXY

In this section, we expand on the theoretical grounding of our framework, specifically elucidating how the Supervised Fine-Tuning loss term in our unified objective (Eq. 3 in our main paper) can be formally interpreted as a dynamic, data-driven Kullback-Leibler (KL) divergence penalty. This perspective is crucial for the understanding of why learning the balance parameter $\mu$ is a more principled approach than using a fixed KL penalty, as is common in traditional RLHF pipelines (Ziegler et al., 2019).

Our goal is to demonstrate that minimizing the SFT loss is mathematically equivalent to minimizing the KL divergence between the expert demonstration distribution, which we denote as $\pi_{\text{demo}}$, and the model's policy, $\pi_\theta$.

**Decomposing the KL Divergence.**   We begin with the formal definition of the KL divergence from the expert distribution $\pi_{\text{demo}}$ to the policy distribution $\pi_\theta$:

$$D_{\text{KL}}(\pi_{\text{demo}}||\pi_\theta) = \mathbb{E}_{(x,y)\sim\pi_{\text{demo}}} \left[ \log \frac{\pi_{\text{demo}}(y|x)}{\pi_\theta(y|x)} \right] \tag{14}$$

By the properties of logarithms, we can decompose this expression into two distinct terms:

$$D_{\text{KL}}(\pi_{\text{demo}}||\pi_\theta) = \underbrace{\mathbb{E}_{(x,y)\sim\pi_{\text{demo}}}[\log \pi_{\text{demo}}(y|x)]}_{\text{Term 1: Negative Entropy of Expert}}$$
$$- \underbrace{\mathbb{E}_{(x,y)\sim\pi_{\text{demo}}}[\log \pi_\theta(y|x)]}_{\text{Term 2: Negative SFT Loss}} \tag{15}$$

**Analyzing the Components.**   We now analyze each term in Eq. 15 with respect to the optimization of the model parameters $\theta$:

- **Term 2** is precisely the expectation of the log-likelihood of the demonstration data under the model's policy. This is, by definition, the negative of the standard SFT loss function:
$$-\mathbb{E}_{(x,y)\sim\pi_{\text{demo}}}[\log \pi_\theta(y|x)] = L_{\text{SFT}}(\theta) \tag{16}$$

- **Term 1** is the negative entropy of the expert demonstration distribution, $-H(\pi_{\text{demo}})$. The crucial insight here is that the expert distribution $\pi_{\text{demo}}$ is defined by the static, pre-existing SFT dataset ($\mathcal{D}_{\text{SFT}}$). Therefore, its entropy is a fixed constant value with respect to the model parameters $\theta$ that we are optimizing.

**Establishing Equivalence.** Substituting these observations back into Eq. 15, we get:

$$D_{\text{KL}}(\pi_{\text{demo}}||\pi_\theta) = -H(\pi_{\text{demo}}) + L_{\text{SFT}}(\theta) \tag{17}$$

This reveals a direct linear relationship between the KL divergence and the SFT loss. When we seek to find the optimal parameters $\theta^*$ that minimize the SFT loss, we are performing the following optimization:

$$\theta^*_{\text{SFT}} = \arg\min_\theta L_{\text{SFT}}(\theta)$$
$$= \arg\min_\theta \left( D_{\text{KL}}(\pi_{\text{demo}}||\pi_\theta) + H(\pi_{\text{demo}}) \right) \tag{18}$$

Since $H(\pi_{\text{demo}})$ is a constant with respect to $\theta$, minimizing the SFT loss is mathematically equivalent to minimizing the KL divergence from the expert distribution to the policy distribution.

**Implication for AMFT: A Principled, Adaptive Regularizer.** This theoretical connection provides a deeper justification for the AMFT framework. The SFT term, weighted by $\mu$, in our unified loss function:

$$L_{\text{total}}(\theta; \mu) = (1 - \mu) \cdot L_{\text{RL}}(\theta) + \mu \cdot L_{\text{SFT}}(\theta) \tag{19}$$

is not merely an imitation objective. It functions as a **principled KL divergence penalty** that regularizes the policy $\pi_\theta$ against deviating too far from the expert demonstration distribution $\pi_{\text{demo}}$.

This contrasts sharply with the fixed KL penalty commonly used in RLHF, which typically regularizes against the base model $\pi_{\text{ref}}$. While regularizing against $\pi_{\text{ref}}$ prevents catastrophic forgetting of pre-trained knowledge, it can be overly restrictive, penalizing the model for learning novel, desirable behaviors present in the expert data.

In AMFT, the meta-gradient controller learns the optimal, time-varying schedule for the weight $\mu$. From this theoretical standpoint, the controller is effectively learning the optimal Lagrange multiplier for the imitation constraint ($D_{\text{KL}}(\pi_{\text{demo}}||\pi_\theta)$) at each stage of training. It learns when to strongly pull the policy towards the expert distribution (high $\mu$) for stability and structured reasoning, and when to relax this constraint (low $\mu$) to allow for reward-driven exploration. This makes AMFT's approach to balancing imitation and exploration more principled, dynamic, and directly tied to the optimization landscape than methods relying on fixed KL penalties or reactive heuristics.

### B.3 Assumptions and Limitations of the Meta-Gradient Approximation

The computational tractability of our meta-gradient controller hinges on a one-step approximation of the inner optimization loop, a technique well-established in the meta-learning literature (Franceschi et al., 2018). While powerful, this approach rests on several key assumptions and entails certain limitations that are important to acknowledge for a complete understanding of AMFT's behavior. This section details these theoretical and practical trade-offs.

**The Core Assumption: A One-Step Lookahead as a Valid Proxy.** The fundamental assumption of our approximation is that the effect of a change in the balancing weight $\mu$ on the model parameters $\theta$ after a single gradient descent step is a sufficiently informative proxy for its long-term impact. As detailed in Appendix C.2, we approximate the full Jacobian $\frac{\partial \theta_t}{\partial \mu_t}$ by considering only the most recent update: $\theta_t \approx \theta_{t-1} - \alpha_\theta \nabla_\theta L_{\text{total}}(\theta_{t-1}; \mu_t)$. This assumption holds most reliably under two conditions: (1) a sufficiently small inner-loop learning rate $\alpha_\theta$, which ensures that single updates do not drastically alter the loss landscape, and (2) a relatively smooth loss landscape for $L_{\text{total}}$ with respect to $\theta$. If the landscape were highly chaotic, a single gradient step would not be representative of the optimization trajectory, and the one-step lookahead would provide a noisy and unreliable signal for updating $\mu$. Our empirical results suggest that for the LLM fine-tuning scenarios we study, these conditions are adequately met to allow for stable meta-optimization.

**Limitation I: Inherent Myopia and Long-Term Credit Assignment.** A direct consequence of the one-step approximation is its inherent **myopia**. The meta-gradient is calculated based on the immediate, next-step impact on the validation utility $U(\theta)$. It cannot capture complex, long-term dependencies where a particular choice for $\mu$ might lead to a temporary dip in validation performance but unlock a more promising optimization trajectory several steps later. This represents a classic

trade-off between computational feasibility and optimization foresight. While a full unrolling of the optimization history would provide a more accurate gradient for $\mu$, its computational and memory costs are prohibitive. AMFT's design acknowledges this trade-off, using the efficient one-step approximation to provide a principled, forward-looking signal that, while not perfectly prescient, is a significant advance over the purely reactive signals used in prior work.

**Limitation II: Dependence on the Validation Set.** The quality and representativeness of the validation set, $\mathcal{D}_{val}$, are critical to the success of the meta-learning controller. The meta-gradient $\nabla_\mu U(\theta_t)$ is computed with respect to performance on this set, meaning the learned schedule for $\mu$ will be optimized to produce a model that excels specifically on data distributed similarly to $\mathcal{D}_{val}$. If the validation set is small, noisy, or poorly aligned with the final test distribution, the controller may learn a suboptimal or even detrimental schedule for $\mu$. This highlights the importance of curating a high-quality, representative validation set, a prerequisite shared by many meta-learning and hyperparameter optimization techniques.

**Limitation III: Potential for Instability and the Role of the Entropy Heuristic.** The meta-gradient calculation involves second-order information (as seen in the Hessian-vector products implicitly computed when differentiating through a gradient step). In optimization landscapes that are not sufficiently smooth, these higher-order derivatives can be noisy, potentially introducing instability into the updates for $\mu$. This is a primary motivation for the hybrid nature of our adaptive weight controller. The short-term, reactive entropy heuristic ($\eta_H(H^* - H(\pi_{\theta_t}))$ in Eq. 6) acts as a crucial **regularizer and stabilizer**. It provides a fast-acting, robust signal based on the immediate stability of the policy (its entropy), complementing the long-term, forward-looking (but potentially noisy) meta-gradient. Our ablation study in Table 4 of the main paper, which shows a significant performance drop when removing this heuristic ('w/o Entropy Heuristic'), empirically validates its critical role in ensuring a stable and effective learning process.

Despite these limitations, the one-step meta-gradient provides a principled, forward-looking optimization signal for the imitation-exploration balance, representing a significant advance over the purely reactive, heuristic-based mechanisms employed in prior adaptive frameworks.

## C AMFT Algorithm Implementation Details

This section provides a detailed, step-by-step description of the AMFT algorithm, designed to supplement the high-level overview in the main paper and facilitate replication. Our goal is to offer a transparent and comprehensive guide to the implementation of the core mechanisms, including the SFT warm-up, the main adaptive training loop, and the meta-gradient-based weight controller.

### C.1 Fully Annotated Pseudocode

Algorithm 2 presents the complete, annotated pseudocode for Adaptive Meta Fine-Tuning (AMFT). To offer a clear and in-depth understanding of its operational flow, we first elaborate on the rationale behind its three-phase structure. This design is intentionally crafted to address the well-documented instabilities of pure RL and the catastrophic forgetting issues of sequential SFT→RL pipelines (Chen et al., 2025; Fernando et al., 2025).

---

**Algorithm 2** The AMFT Algorithm (Detailed Version)

---

**Require:** Pretrained model policy $\pi_\theta$ with parameters $\theta$.
**Require:** Value function $V_\phi$ with parameters $\phi$.
**Require:** SFT demonstration dataset $\mathcal{D}_{\text{SFT}}$.
**Require:** Validation dataset for meta-objective $\mathcal{D}_{\text{val}}$.
**Require:** Environment env with reward function $R_{\text{explicit}}$.
**Require:** Hyperparameters: SFT warm-up steps $W$, total training steps $T$, initial weight $\mu_{\text{init}}$, controller learning rates $\eta_\mu, \eta_H$, policy learning rate $\alpha_\theta$, value function learning rate $\alpha_\phi$, meta-update frequency $K$, weight clip range $[\mu_{\min}, \mu_{\max}]$.
**Ensure:** Fine-tuned model policy $\pi_\theta^*$.

1: **Initialization:**
2: Initialize policy parameters $\theta_0$ from pretrained model.
3: Initialize value function parameters $\phi_0$.
4: Initialize adaptive weight $\mu_0 \leftarrow \mu_{\text{init}}$.
5: *// — Phase 1: SFT Warm-up —*
6: **for** $w = 1$ **to** $W$ **do**
7:     Sample a batch $\{(x_i, y_i)\}_{i=1}^m \sim \mathcal{D}_{\text{SFT}}$.
8:     Compute SFT loss $L_{\text{SFT}}(\theta_{w-1})$ using Eq. 2 (main paper).
9:     Update policy parameters: $\theta_w \leftarrow \theta_{w-1} - \alpha_\theta \nabla_\theta L_{\text{SFT}}(\theta_{w-1})$.
10: **end for**
11: Compute target entropy $H^* \leftarrow \text{mean}_{x \sim \mathcal{D}_{\text{SFT}}} H(\pi_{\theta_W}(\cdot|x))$. {Set target entropy based on post-SFT policy.}
12: Initialize main loop policy $\theta_0 \leftarrow \theta_W$.
13: *// — Phase 2: Main Adaptive Training Loop —*
14: **for** $t = 0$ **to** $T - 1$ **do**
15:     *// — Data Collection —*
16:     Sample a batch of demonstrations $\{(x_i, y_i)\}_{i=1}^m \sim \mathcal{D}_{\text{SFT}}$.
17:     Generate a batch of on-policy rollouts $\{\tau_j\}_{j=1}^n \sim \pi_{\theta_t}$ in env.
18:     *// — Loss Computation —*
19:     Compute $L_{\text{SFT}}(\theta_t)$ on the SFT batch.
20:     Compute rewards, advantages, and $L_{\text{RL}}(\theta_t)$ on the RL rollouts.
21:     *// — Adaptive Weight Controller Update —*
22:     $g_\mu \leftarrow 0$.
23:     **if** $t \pmod K == 0$ **then**
24:         Compute meta-gradient $g_\mu \leftarrow \nabla_\mu U(\theta_t)$ on a validation batch. {Meta-gradient is updated periodically. See Appendix B.2 for the full derivation.}
25:     **end if**
26:     Compute policy entropy $H_t \leftarrow \text{mean}_{x \sim \mathcal{D}_{\text{SFT}}} H(\pi_{\theta_t}(\cdot|x))$.
27:     Compute entropy heuristic $g_H \leftarrow H^* - H_t$.
28:     Update adaptive weight using Eq. 6 (main paper):
29:     $\mu_{t+1} \leftarrow \text{clip}(\mu_t + \eta_\mu g_\mu + \eta_H g_H, \mu_{\min}, \mu_{\max})$.
30:     *// — Policy and Value Function Update —*
31:     Compute the unified loss using Eq. 3 (main paper):
32:     $L_{\text{total}}(\theta_t; \mu_{t+1}) \leftarrow (1 - \mu_{t+1}) \cdot L_{\text{RL}}(\theta_t) + \mu_{t+1} \cdot L_{\text{SFT}}(\theta_t)$.
33:     Update policy parameters: $\theta_{t+1} \leftarrow \theta_t - \alpha_\theta \nabla_\theta L_{\text{total}}(\theta_t; \mu_{t+1})$.
34:     Update value function parameters $\phi_{t+1}$ using its own loss (e.g., MSE on returns from rollouts).
35: **end for**
36: **return** $\pi_{\theta_T}$

---

**SFT Warm-up.** The algorithm begins with a mandatory, albeit brief, warm-up phase consisting solely of Supervised Fine-Tuning. This stage is critical for two primary reasons. First, it serves as a "format teacher" (Zhou et al., 2024), aligning the base model's outputs with the required structural conventions (e.g., reasoning chains, answer formats). This initial alignment is essential for the subsequent RL phase, as it ensures that the policy can generate syntactically valid trajectories from which a meaningful reward signal can be extracted. This directly mitigates the "advantage collapse" problem, where a chaotic initial policy fails to produce any reward-bearing outputs, leading to a null learning signal (Liu et al., 2025a). Second, this phase establishes a stable and competent policy

whose average token entropy can serve as a reliable target, $H^*$, for the entropy-based heuristic in our adaptive weight controller. This provides a data-driven anchor for what constitutes a "stable" level of policy uncertainty.

**Main Adaptive Training Loop.** Following the warm-up, AMFT transitions to its core single-stage training loop. Unlike sequential methods that create a hard switch between objectives, this loop continuously integrates both imitation and exploration signals. At each step, the algorithm processes a mixed batch of data, comprising both expert demonstrations from $\mathcal{D}_{\text{SFT}}$ and on-policy rollouts generated by the current policy. The central innovation lies in how these two signals are balanced: the adaptive weight controller (lines 8-11 in the pseudocode) updates the balancing parameter $\mu$ *before* it is used to compute the unified loss for the policy update. This ensures that every gradient step taken by the policy is guided by the most current, principled assessment of the optimal imitation-exploration trade-off. The controller itself synergizes a forward-looking meta-gradient with a reactive entropy-based regularizer, a mechanism designed to pursue long-term performance while maintaining short-term stability. This stands in contrast to prior heuristic-based methods that rely solely on reactive signals like reward density or gradient norms (Liu et al., 2025c; Chen et al., 2025).

## C.2 META-GRADIENT APPROXIMATION FOR THE ADAPTIVE WEIGHT CONTROLLER

A central innovation of AMFT is its ability to learn the optimal SFT-RL balance rather than relying on heuristics. This is achieved by treating the balancing weight $\mu$ as a learnable parameter, which is updated via meta-learning. This section provides a detailed derivation of the one-step meta-gradient approximation used in our adaptive weight controller, as first introduced by (Franceschi et al., 2018).

**Problem Formulation: A Bilevel Optimization.** The core task is a bilevel optimization problem. In the **inner loop**, we update the policy parameters $\theta$ to minimize the unified loss $L_{\text{total}}$ for a *given* weight $\mu$. In the **outer loop**, we aim to update $\mu$ to maximize a long-term utility function $U(\theta)$ evaluated on a separate validation set.

Let $\theta_{t-1}$ be the policy parameters before an update. The inner-loop update for one step is:

$$\theta_t(\mu_t) = \theta_{t-1} - \alpha_\theta \nabla_\theta L_{\text{total}}(\theta_{t-1}; \mu_t) \tag{20}$$

where we explicitly denote $\theta_t$ as a function of $\mu_t$ to highlight the dependency. The outer-loop objective is the validation performance:

$$U(\theta_t) = \mathbb{E}_{(x,\tau) \sim \pi_{\theta_t}(\cdot | \mathcal{D}_{\text{val}})}[R_{\text{explicit}}(\tau)] \tag{21}$$

Our goal is to compute the meta-gradient $\nabla_\mu U(\theta_t)$, which tells us how to adjust $\mu_t$ to maximize this long-term utility.

**The Meta-Gradient via the Chain Rule.** Using the chain rule, the gradient of the outer-loop objective with respect to $\mu_t$ is:

$$\nabla_\mu U(\theta_t) = \underbrace{\nabla_\theta U(\theta_t)}_{\text{Outer Gradient}} \cdot \underbrace{\frac{\partial \theta_t}{\partial \mu_t}}_{\text{Jacobian}} \tag{22}$$

This equation decomposes the meta-gradient into two components:

- **The Outer Gradient** ($\nabla_\theta U(\theta_t)$): This is the gradient of the validation utility with respect to the *updated* model parameters $\theta_t$. It indicates the direction in parameter space that improves validation performance.

- **The Jacobian** ($\frac{\partial \theta_t}{\partial \mu_t}$): This term captures how the model parameters $\theta_t$ change in response to an infinitesimal change in the balancing weight $\mu_t$.

Computing the full Jacobian is computationally prohibitive, as it requires unrolling the entire training history to account for how $\mu_t$ influences all subsequent parameter updates.

**The One-Step Approximation.** To make this computation tractable, we adopt a widely-used one-step approximation from the meta-learning literature (Franceschi et al., 2018). We approximate the updated parameters $\theta_t$ using only the single, most recent gradient descent step from Eq. 20. We can then differentiate this one-step update rule with respect to $\mu_t$:

$$\frac{\partial \theta_t}{\partial \mu_t} \approx \frac{\partial}{\partial \mu_t} \left(\theta_{t-1} - \alpha_\theta \nabla_\theta L_{\text{total}}(\theta_{t-1}; \mu_t)\right)$$

$$= -\alpha_\theta \frac{\partial}{\partial \mu_t} \nabla_\theta L_{\text{total}}(\theta_{t-1}; \mu_t) \tag{23}$$

Since $\theta_{t-1}$ does not depend on the *current* weight $\mu_t$, its derivative is zero. Now, we substitute the definition of $L_{\text{total}} = (1 - \mu_t)L_{\text{RL}} + \mu_t L_{\text{SFT}}$:

$$\frac{\partial \theta_t}{\partial \mu_t} \approx -\alpha_\theta \frac{\partial}{\partial \mu_t} \nabla_\theta \left[(1 - \mu_t)L_{\text{RL}}(\theta_{t-1}) + \mu_t L_{\text{SFT}}(\theta_{t-1})\right]$$

$$= -\alpha_\theta \nabla_\theta \frac{\partial}{\partial \mu_t} \left[(1 - \mu_t)L_{\text{RL}}(\theta_{t-1}) + \mu_t L_{\text{SFT}}(\theta_{t-1})\right]$$

$$= -\alpha_\theta \nabla_\theta \left[-L_{\text{RL}}(\theta_{t-1}) + L_{\text{SFT}}(\theta_{t-1})\right]$$

$$= -\alpha_\theta \left(\nabla_\theta L_{\text{SFT}}(\theta_{t-1}) - \nabla_\theta L_{\text{RL}}(\theta_{t-1})\right) \tag{24}$$

Here, we can swap the order of differentiation because the losses $L_{\text{RL}}$ and $L_{\text{SFT}}$ are evaluated at $\theta_{t-1}$, which is constant with respect to $\mu_t$.

**The Final Meta-Gradient Formula and its Intuition.** By substituting this tractable approximation of the Jacobian back into the chain rule (Eq. 22), we obtain the final formula for the meta-gradient:

$$\nabla_\mu U(\theta_t) \approx -\alpha_\theta \nabla_\theta U(\theta_t)^\top \left(\nabla_\theta L_{\text{SFT}}(\theta_{t-1}) - \nabla_\theta L_{\text{RL}}(\theta_{t-1})\right) \tag{25}$$

This final expression is computationally efficient, requiring only three gradient calculations: one on a validation batch for the outer gradient, and two on the training batch for the SFT and RL gradients (which are already computed for the main policy update).

The intuition behind this formula is powerful. The term $(\nabla_\theta L_{\text{SFT}} - \nabla_\theta L_{\text{RL}})$ represents the "disagreement vector" in the parameter space between the imitation and exploration objectives. The meta-gradient is the projection of the validation performance gradient ($\nabla_\theta U$) onto this disagreement vector.

- If increasing $\mu$ (i.e., moving more towards the SFT gradient direction) would result in new parameters $\theta_t$ that better align with the direction of long-term improvement $\nabla_\theta U(\theta_t)$, the meta-gradient will be positive. This will increase $\mu$ in the next step, favoring SFT.
- Conversely, if moving more towards the RL direction (decreasing $\mu$) better aligns with long-term improvement, the meta-gradient will be negative, thus decreasing $\mu$ and favoring RL.

This mechanism is explicitly **forward-looking**: the decision on how to balance SFT and RL at step $t$ is based on its approximated effect on performance at a future step. This allows AMFT to learn a principled training curriculum that directly optimizes for the final task objective, moving beyond the reactive, proxy-based heuristics of prior work.

# D EXPERIMENTAL SETUP

This section provides a comprehensive overview of the experimental configurations used to evaluate AMFT and all baseline methods. Our goal is to ensure full reproducibility by detailing the datasets, models, hyperparameters, and evaluation protocols.

## D.1 DATASET DETAILS

Our evaluation spans three distinct reasoning domains: mathematical reasoning, general reasoning (for out-of-distribution testing), and visual reasoning. The selection of these datasets is intended to provide a rigorous and multi-faceted assessment of each fine-tuning paradigm's ability to foster both specialized competence and broad generalization.

**Mathematical Reasoning Datasets (In-Distribution).** For fine-tuning and in-distribution evaluation of mathematical reasoning, we use a combination of a primary training dataset and five standard evaluation benchmarks.

- **Training Dataset (OpenR1-Math-46k-8192):** As stated in the main paper, all math-focused fine-tuning originates from this dataset (Yan et al., 2025). It consists of 46,000 mathematical problems with high-quality, step-by-step reasoning solutions (CoT) generated by the DeepSeek-R1 model. This dataset serves a dual purpose in our framework: its problem statements are used as prompts for RL rollouts, and its detailed solutions are used as expert demonstrations for the SFT objective.

- **Evaluation Benchmarks:** We evaluate performance on five challenging, competition-level mathematics benchmarks to measure in-domain reasoning capabilities: **AIME24** and **AMC** (Li et al., 2024) for competitive math, **MATH500** (Hendrycks et al., 2021) and **Minerva** (Lewkowycz et al., 2022) for broad mathematical problem-solving, and **OlympiadBench** (He et al., 2024) for problems requiring exceptional insight.

**General Reasoning Datasets (Out-of-Distribution).** To assess how well mathematical reasoning skills generalize to other knowledge-intensive domains, we evaluate all models on three OOD benchmarks. These tasks require reasoning but do not fall into the domain of pure mathematics, thus testing for catastrophic forgetting of general knowledge.

- **ARC-C** (Clark et al., 2018): The AI2 Reasoning Challenge (Challenge set), consisting of difficult, grade-school-level science questions that require commonsense and scientific reasoning.

- **GPQA-D** (Rein et al., 2024): The "Diamond" subset of the Graduate-Level Google-Proof Q&A benchmark, containing expert-level questions in biology, physics, and chemistry designed to be resistant to simple web searches.

- **MMLU-Pro** (Wang et al., 2024): A more robust and challenging version of the Massive Multitask Language Understanding benchmark, covering 57 diverse subjects and designed to test deep knowledge and its application.

**Visual Reasoning Datasets and Generalization Splits.** For visual reasoning, we use the General Points and V-IRL benchmarks, which are specifically designed to test generalization across both textual rule changes and visual variations. We follow the experimental design of (Chu et al., 2025) to define our ID and OOD splits.

- **General Points:** An arithmetic reasoning task where the model is presented with four playing cards (either as text or an image) and must generate a mathematical expression that equals a target number (24).
  - **ID vs. OOD (Rule Variation):** This split tests the model's ability to apply arithmetic principles under changing rules.
    * *In-Distribution (ID):* Models are trained and evaluated using the rule where face cards 'J', 'Q', and 'K' all count as the number **10**.
    * *Out-of-Distribution (OOD):* Models are evaluated on an unseen rule where 'J', 'Q', and 'K' are interpreted as **11, 12, and 13**, respectively. This forces the model to generalize its arithmetic operations rather than memorize specific number combinations.
  - **ID vs. OOD (Visual Variation):** This split tests the model's visual recognition capabilities, specifically its invariance to cosmetic features like color.
    * *In-Distribution (ID):* Models are trained exclusively on images of cards with **black suits** (spades ♠ and clubs ♣).
    * *Out-of-Distribution (OOD):* Models are evaluated on images of cards with **red suits** (hearts ♡ and diamonds ◇).

- **V-IRL (Vision-Language Navigation):** A spatial reasoning task where the model must navigate a real-world environment based on visual observations and textual instructions.
  - **ID vs. OOD (Rule Variation):** This split tests whether the model learns abstract navigational concepts or memorizes a specific action vocabulary.

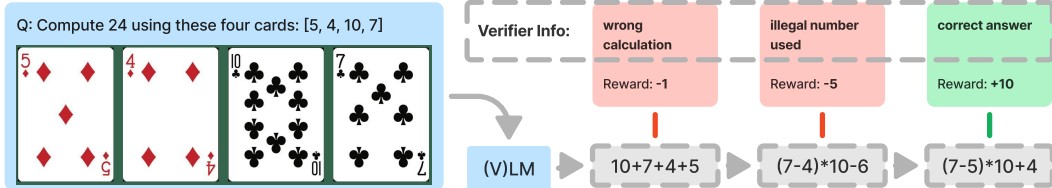

Figure 4: **General Points:** An example of the sequential revision formulation with a verifier. The illustration is from (Chu et al., 2025).

* *In-Distribution (ID):* Models are trained using an **absolute orientation** action space (e.g., 'turn_direction(north)', 'turn_direction(west)') .
* *Out-of-Distribution (OOD):* Models are evaluated using a **relative orientation** action space (e.g., 'turn_direction(left)', 'turn_direction(slightly right)').

– **ID vs. OOD (Visual Variation):** This split tests the model's ability to generalize its spatial reasoning and landmark recognition to novel environments.

* *In-Distribution (ID):* Models are trained on navigation routes collected exclusively from **New York City**.
* *Out-of-Distribution (OOD):* Models are evaluated on the VLN mini benchmark, which contains routes from various other cities worldwide, such as **Milan, London, and Hong Kong**.

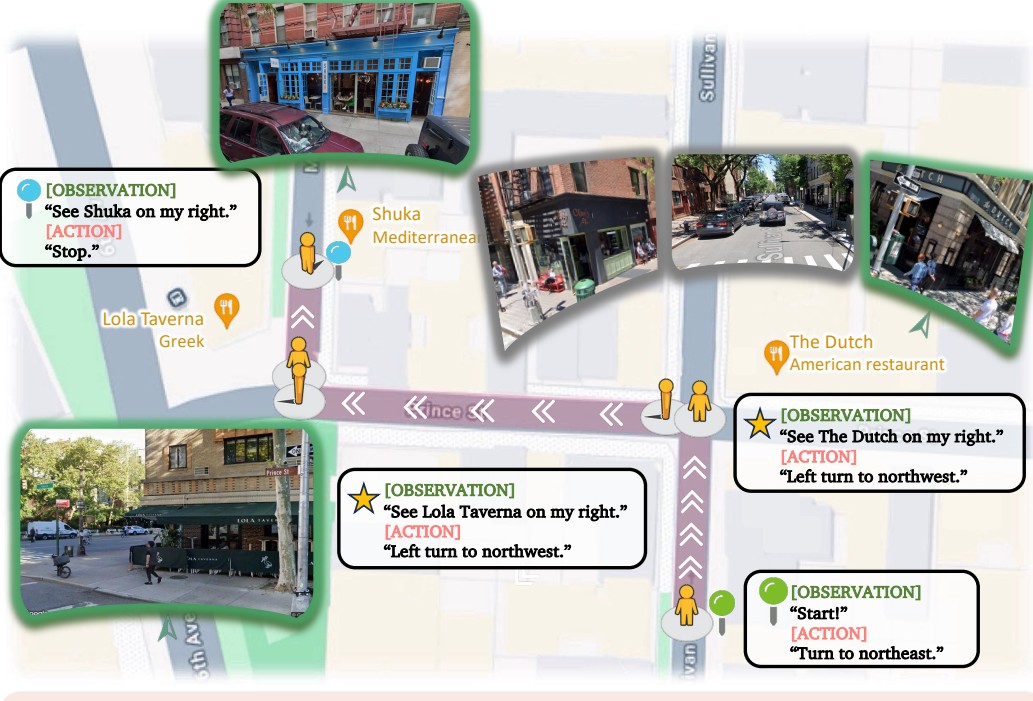

Figure 5: **V-IRL:** Demonstration of one navigation task. The navigation procedure is shown at the top, with the navigation instructions displayed below. Visual observation-related information is highlighted in green, while action-related information is marked in orange. The illustration is from (Chu et al., 2025).

## D.2   MODEL DETAILS

The selection of base models is a critical component of our experimental design, chosen to rigorously test the AMFT framework's effectiveness across both specialized and general-purpose reasoning domains. We provide detailed descriptions of the models used for mathematical and visual reasoning tasks below. Our choice of these specific, publicly-available models ensures that our results are transparent and reproducible.

**Qwen2.5-Math-7B for Mathematical Reasoning.**   For all mathematical reasoning experiments, we use **Qwen2.5-Math-7B** as the base model (Yang et al., 2024a). This model is part of the Qwen2.5 series developed by Alibaba and has been specifically optimized for mathematical tasks through continued pre-training on extensive math-related corpora (Yan et al., 2025; Cui et al., 2025a). Its strong foundational capabilities in arithmetic, algebra, and logic make it an ideal and challenging baseline for evaluating advanced fine-tuning methods. By starting with a model that already possesses strong innate reasoning abilities, we can more accurately assess the additional value and sample efficiency provided by our AMFT paradigm compared to other state-of-the-art fine-tuning techniques.

**LLaMA-3.2-Vision-11B for Visual Reasoning.**   For the multi-modal reasoning tasks, we employ **LLaMA-3.2-Vision-11B** (AI@Meta, 2024). This model is a state-of-the-art, open-source Vision-Language Model (VLM) developed by Meta. It is known for its robust visual understanding and strong instruction-following capabilities, which are essential prerequisites for tackling complex, multi-step visual reasoning problems (Chu et al., 2025). The model's 11-billion-parameter scale provides sufficient capacity for the nuanced demands of both spatial navigation (V-IRL) and symbolic visual reasoning (General Points). Using this powerful, general-purpose VLM allows us to test AMFT's ability to resolve the "SFT Memorizes, RL Generalizes" dilemma in a multi-modal context where both visual perception and logical deduction are intertwined.

## D.3   BASELINE IMPLEMENTATION DETAILS

To rigorously evaluate the performance of AMFT, we established a comprehensive suite of baseline methods. This section provides a detailed description of the implementation and hyperparameter configurations for each baseline. Our goal is to ensure a fair and transparent comparison by not only listing the parameters but also explaining the rationale behind their selection, grounding our choices in established best practices from the cited literature. All experiments were conducted using the same underlying computational framework and base models as AMFT to isolate the effects of the training paradigm itself.

**SFT-only.**   This baseline represents the standard supervised fine-tuning paradigm. It serves to quantify the effectiveness of pure imitation learning on the reasoning datasets and acts as the foundational first stage for the sequential SFT→RL pipeline.

- **Objective and Rationale:** The model is trained exclusively by minimizing the standard cross-entropy loss (negative log-likelihood) on the high-quality demonstration data from the respective training sets ($\mathcal{D}_{\text{SFT}}$). The objective is to directly imitate the expert policy ($\pi_{\text{demo}}$) encoded in the dataset. This approach is foundational for teaching the model domain-specific knowledge and, critically, the structural and stylistic patterns of the desired reasoning format (Zhou et al., 2024). As observed by (Chu et al., 2025), this initial format teaching is often a prerequisite for successful RL.

- **Implementation Details:** We conducted full-parameter fine-tuning to provide the model with maximum flexibility to adapt to the complex reasoning structures present in the data. The training was run for 3 epochs, a standard duration found in related works to achieve a good balance between sufficient exposure to the data and the risk of overfitting on large-scale instruction datasets (Yan et al., 2025).

- **Hyperparameter Configuration:** The chosen hyperparameters reflect common practices for effective SFT.
  - **Learning Rate** ($5 \times 10^{-5}$)**:** This is a conventional, relatively high learning rate for SFT, designed for rapid adaptation of a large pre-trained model to a new data distribution. This contrasts sharply with the much lower learning rates required for stable RL, a key difference between the optimization dynamics of the two paradigms (Rajani et al., 2025).

– **Scheduler (Cosine Annealing):** We used a cosine learning rate scheduler with a warm-up ratio of 0.1. This allows for stable initial updates, followed by a smooth decay of the learning rate, which has been empirically shown to improve convergence and lead to more robust final models.

**RL-only (GRPO).**    This baseline is designed to isolate the effect of reinforcement learning by applying it directly to the base model, reflecting the "RL from scratch" or "zero RL" paradigm (Zeng et al., 2025). It is a critical baseline for assessing whether RL can instill reasoning abilities without a supervised warm-up.

- **Objective and Rationale:** We employ Group Relative Policy Optimization (GRPO), a state-of-the-art policy gradient algorithm for RLVR popularized by (Shao et al., 2024). GRPO is particularly well-suited for reasoning tasks as it computes advantages by normalizing rewards within a group of self-generated trajectories for a given prompt. This eliminates the need for a separate, learned critic network, reducing algorithmic complexity and computational overhead. The policy is optimized solely based on the explicit, outcome-based reward signal $R_{\text{explicit}}$.

- **Implementation Details:** The model learns exclusively from on-policy rollouts. At each step, 8 candidate trajectories are generated per prompt to form the group for advantage calculation. The model is trained for a fixed 500 optimization steps to maintain a consistent computational budget across all RL-based comparisons.

- **Hyperparameter Configuration:** The RL hyperparameters are chosen with a strong emphasis on training stability.

  – **Learning Rate ($1 \times 10^{-6}$):** A very low learning rate is crucial for the stability of on-policy RL algorithms like GRPO. Higher rates can cause the policy to change too drastically between updates, which violates the assumptions of importance sampling and leads to trust region collapse and divergent training (Rajani et al., 2025).
  – **Discount Factor ($\gamma = 1.0$):** For episodic reasoning tasks with a sparse final reward, there is no need to discount future rewards. A value of 1.0 ensures that the credit for a correct final answer is fully and equally propagated back to all steps in the reasoning trajectory that produced it.

**Sequential SFT→RL.**    This baseline represents the conventional two-stage fine-tuning pipeline, a widely adopted de facto standard for aligning powerful LLMs. It serves to benchmark the benefits and drawbacks of sequential training, particularly the issue of catastrophic forgetting that AMFT aims to solve.

- **Objective and Rationale:** The two-stage process is designed to leverage the distinct strengths of SFT and RL sequentially. Stage 1 (SFT) provides a strong initialization by teaching the model the required reasoning format and aligning it with a distribution of high-quality solutions. This pre-conditioning is vital for making the subsequent RL stage tractable, as it provides a policy already capable of producing some reward-generating trajectories, thus avoiding the severe sample inefficiency and potential collapse of starting RL from a naive policy (Chu et al., 2025). Stage 2 (RL) then refines this policy, exploring variations and optimizing for correctness beyond what is present in the static SFT dataset.

- **Implementation Details:** For maximum fairness and to isolate the effect of the sequential paradigm itself, our implementation is a direct composition of the two preceding baselines.

  1. We first perform the complete **SFT-only** training and select the checkpoint with the highest validation performance.
  2. We then initialize the **RL** phase from this best SFT checkpoint, using the exact same hyperparameters and 500-step training budget as the **RL-only** baseline.

This controlled setup ensures that any performance difference between this baseline and AMFT can be directly attributed to the training paradigm (sequential vs. unified) rather than to differences in initialization or optimization settings.

**Other State-of-the-Art Hybrid Methods.**    To ensure that AMFT's performance is benchmarked against the current state-of-the-art, we implemented several leading single-stage hybrid frameworks.

For each method, we adhered as closely as possible to the methodologies and critical hyperparameter settings described in their original publications. This approach guarantees that our baselines are not just strawman implementations but are faithful, strong representations of these advanced techniques.

- **LUFFY** (Yan et al., 2025): This method enhances on-policy RLVR with off-policy guidance from stronger models.
    - **Core Mechanism:** LUFFY augments the on-policy rollout batch with high-quality, off-policy expert demonstrations and uses a Mixed-Policy GRPO objective to balance imitation and exploration.
    - **Implementation Details:** In our implementation, for every batch of 8 rollouts per prompt, we used **7 on-policy rollouts** generated by the current policy and **1 off-policy expert trace** from the $\mathcal{D}_{\text{SFT}}$ dataset, as recommended by the authors .
    - **Key Hyperparameters:**
        * The off-policy guidance is integrated by setting the behavior policy probability $\pi_\phi = 1$ for computational efficiency, which avoids tokenization mismatches and the need to re-compute probabilities for the expert model .
        * The PPO-clip operation was omitted for the off-policy objective, as the standard clipping becomes imbalanced when $\pi_\phi = 1$ .
        * The policy shaping function $f(x) = x/(x + \gamma)$ was applied to the off-policy importance sampling ratio, with the hyperparameter $\gamma$ set to **0.1**, the optimal value found in their ablation studies .

- **ReLIFT** (Ma et al., 2025a): This approach is characterized by its strategy of interleaving RL with targeted online fine-tuning.
    - **Core Mechanism:** ReLIFT alternates between standard RL steps and SFT steps. Crucially, the SFT updates are performed specifically on expert demonstrations corresponding to thehardest questions—those for which the policy failed to generate a correct response during recent rollouts.

- **TAPO** (Wu et al., 2025): This framework enhances RL by incorporating high-level, structured guidance in the form of thought patterns.
    - **Core Mechanism:** TAPO abstracts reasoning strategies (thought patterns) from successful demonstrations and adaptively integrates this external knowledge into the GRPO framework. This provides a higher-level form of guidance than raw token-level imitation.

- **SRFT** (Fu et al., 2025): This method proposes a single-stage, unified loss function that balances SFT and RL signals using policy entropy as a dynamic indicator.
    - **Core Mechanism:** SRFT does not switch between SFT and RL but rather combines their losses in every step, with weights that are dynamically adjusted based on the current policy's entropy, $H(\pi_\theta)$.
    - **Key Hyperparameters:** The weights for the different loss components were calculated dynamically at each step according to the authors' formulations:
        * For SFT loss on demonstration data: $w_{\text{SFT}} = 0.5 \cdot \exp(-H(\pi_\theta))$ . This gives more weight to imitation when the policy is uncertain (high entropy).
        * For RL loss on positive-reward self-generated samples: $w_{\text{RL}} = 0.1 \cdot \exp(H(\pi_\theta))$ . This encourages exploration when the policy becomes too deterministic (low entropy).

By meticulously implementing these diverse and powerful baselines according to their original designs, we provide a robust and challenging context in which to evaluate the unique contributions and superior performance of our AMFT framework.

## D.4 AMFT Hyperparameter Settings

This section provides a comprehensive specification of the hyperparameters used for all AMFT experiments and key baselines discussed in the main paper. Our goal is to ensure full reproducibility and provide clarity on the configurations that led to the reported results. The parameters were chosen based on a combination of preliminary experiments, established best practices from the cited literature, and the specific requirements of our proposed algorithm. We have maintained consistent settings across all comparable methods to ensure a fair and rigorous evaluation.

**General Training and Model Parameters.**  Table 6 details the general hyperparameters applied across all training paradigms, including SFT, RL, and our AMFT framework. These settings relate to the model architecture, optimizer, and learning schedule, and are aligned with established practices for training large reasoning models.

Table 6: General training hyperparameters for all experiments.

| Hyperparameter | Value and Rationale |
|---|---|
| Base Models | Qwen2.5-Math-7B (Math), LLaMA-3.2-Vision-11B (Visual) |
| Optimizer | AdamW |
| AdamW $\beta_1$ | 0.9 |
| AdamW $\beta_2$ | 0.95 |
| AdamW $\epsilon$ | $1 \times 10^{-8}$ |
| Weight Decay | 0.1 |
| Policy Learning Rate ($\alpha_\theta$) | $1 \times 10^{-6}$ |
| Value Function Learning Rate ($\alpha_\phi$) | $5 \times 10^{-6}$ |
| Learning Rate Scheduler | Cosine annealing with warmup |
| Warmup Ratio | 0.1 |
| Total Training Steps ($T$) | 500 (for RL-based methods) |
| SFT Warm-up Steps ($W$) | 50 |

**AMFT Adaptive Weight Controller Parameters.**  The core novelty of AMFT lies in its meta-learning controller, which introduces a new set of hyperparameters. These parameters, detailed in Table 7, govern the behavior of the adaptive weight $\mu$. They were tuned on a small, held-out portion of the training data to find a configuration that yields both stability and strong performance. The use of policy entropy as a stabilizing signal is inspired by the analysis in related works, which identify entropy as a crucial indicator of training effectiveness.

Table 7: Hyperparameters for the AMFT adaptive weight controller.

| Hyperparameter | Value and Rationale |
|---|---|
| Initial Weight ($\mu_{\text{init}}$) | 0.5 (Neutral starting point, balancing SFT and RL) |
| Weight Clip Range $[\mu_{\text{min}}, \mu_{\text{max}}]$ | [0.05, 0.95] (Prevents either objective from being fully ignored) |
| Meta-Gradient Learning Rate ($\eta_\mu$) | $1 \times 10^{-4}$ (Small rate for stable meta-updates) |
| Entropy Heuristic Learning Rate ($\eta_H$) | $5 \times 10^{-4}$ (Allows faster reaction to policy instability) |
| Meta-Update Frequency ($K$) | Every 20 steps (Balances cost and controller responsiveness) |
| Target Entropy ($H^*$) | Data-driven; set to the average policy entropy after the SFT warm-up phase (see Algorithm 2) |

These detailed configurations, grounded in practices from leading contemporary research, provide a transparent foundation for our experimental results and are intended to facilitate direct replication and further extension of our work by the research community.

## D.5  COMPUTATIONAL INFRASTRUCTURE

This section provides the technical specifications of the computational environment utilized for all experiments presented in this paper. Our goal is to ensure full transparency and facilitate the reproducibility of our results by detailing the hardware, software, and core frameworks that underpinned our research. The described infrastructure was designed to be directly comparable to the high-performance environments used in state-of-the-art research, such as the SRFT study, and was consistently used for training and evaluating AMFT and all baseline models.

**Software Environment.**  To maintain a consistent and reproducible software stack, all experiments were run within a containerized environment. The key software components and their versions are listed below:

• **Operating System:** Ubuntu 22.04.2 LTS.

- **NVIDIA Stack:** CUDA Version 12.2, cuDNN 8.9, and NVIDIA Driver Version 535.104.05.
- **Core Libraries:**
    - PyTorch 2.2.1
    - Transformers 4.41.2
    - TRL 0.8.6
    - Accelerate 0.29.3
    - DeepSpeed 0.14.2
    - vLLM 0.4.1 (for efficient RL rollout generation and evaluation)

**Training Framework and Orchestration.** Our experimental pipeline was built on top of established open-source frameworks to ensure robustness and scalability, directly following the toolchain mentioned in our primary reference study.

- **Primary Framework:** The implementation of our AMFT algorithm and all RL-based baselines was built upon the **verl** framework. This choice was made to align our methodology with the SRFT study, ensuring that differences in performance can be attributed to the algorithmic innovations rather than framework-specific optimizations.
- **Distributed Training:** Multi-GPU and multi-node training across the 64-GPU cluster was orchestrated using **Hugging Face Accelerate** in conjunction with **DeepSpeed**, configured with the ZeRO Stage 3 optimization to efficiently manage memory and scale training.
- **Rollout and Evaluation Engine:** To maximize sample efficiency during the reinforcement learning phase and ensure fast evaluation, on-policy rollouts and final evaluations were conducted using the highly optimized **vLLM** inference server. This practice is consistent with several state-of-the-art RLVR frameworks for its speed and efficient memory management.

## E  ADDITIONAL EXPERIMENTAL RESULTS AND ANALYSIS

This section provides deeper, more granular evidence to support the claims made in the main paper. We extend our analysis of training dynamics to the visual reasoning domains and present qualitative case studies that offer concrete examples of AMFT's superior reasoning capabilities compared to baseline methods.

### E.1  TRAINING DYNAMICS VISUALIZATIONS FOR ALL DOMAINS

The main paper (Figure 2) illustrates the learning dynamics of AMFT compared to baselines on mathematical reasoning tasks. To demonstrate the cross-modal robustness and consistency of our meta-learning controller, this section provides equivalent visualizations for our two visual reasoning benchmarks: **General Points** and **V-IRL Navigation**. These plots track both task performance and the trajectory of the adaptive weight $\mu$ over the course of training, offering a clear window into how AMFT autonomously learns an effective training curriculum in multi-modal settings.

**General Points (Visual Arithmetic Reasoning).** The General Points task presents a dual challenge: it requires not only accurate visual recognition of card values from an image but also robust symbolic reasoning to construct a valid mathematical expression. Figure 6 visualizes how different fine-tuning paradigms navigate this complex, multi-modal problem space.

**AMFT's Learned Curriculum.** The trajectory of AMFT (solid blue line) exemplifies the effectiveness of its forward-looking optimization. The adaptive weight $\mu$ (red dash-dotted line) orchestrates a clear, data-driven training curriculum. **In the initial phase (approx. 0-150 steps)**, with $\mu$ held at a high value ($\approx 0.9$), training is dominated by the SFT objective. This forces the model to rapidly master the task's rigid output format and imitate fundamental arithmetic patterns from expert data. This imitation-first approach provides a crucial scaffold that prevents the chaotic, low-reward exploration that often plagues pure RL in its early stages, directly mitigating the risk of policy collapse.

As training progresses, the meta-controller, observing consistent performance gains on the validation set, systematically reduces the weight of $\mu$. This transition signifies that the model has acquired

a competent base policy, allowing the optimization focus to safely shift from imitation towards exploration. In this RL-dominant phase, the model moves beyond merely replicating seen solutions. It begins to refine its strategies, learn from its own mistakes through the explicit reward signal, and discover novel solution paths not present in the static SFT dataset. This ability to explore and generalize is what ultimately allows AMFT to break through the performance plateaus observed in imitation-heavy baselines.

**Baseline Performance Analysis.** The limitations of simpler paradigms are clearly illustrated. The **SFT-only** model (dot-dashed green line) learns the task format quickly but its performance saturates at a low level, a classic exhibition of the "SFT memorizes, RL generalizes" dilemma. It successfully imitates the training data's style but fails to learn the underlying, generalizable arithmetic principles. The **RL-only** baseline (dotted orange line) confirms the challenge of exploration without guidance; it learns slowly and exhibits high variance, showcasing the severe sample inefficiency of pure exploration in a complex, sparse-reward environment. The sequential **SFT→RL** approach (dashed purple line) is a strong baseline, leveraging the SFT initialization to achieve respectable performance. However, it is ultimately surpassed by AMFT, suggesting that the hard switch between training stages is suboptimal compared to AMFT's continuous, adaptive fusion of the two complementary learning signals, which better mitigates catastrophic forgetting.

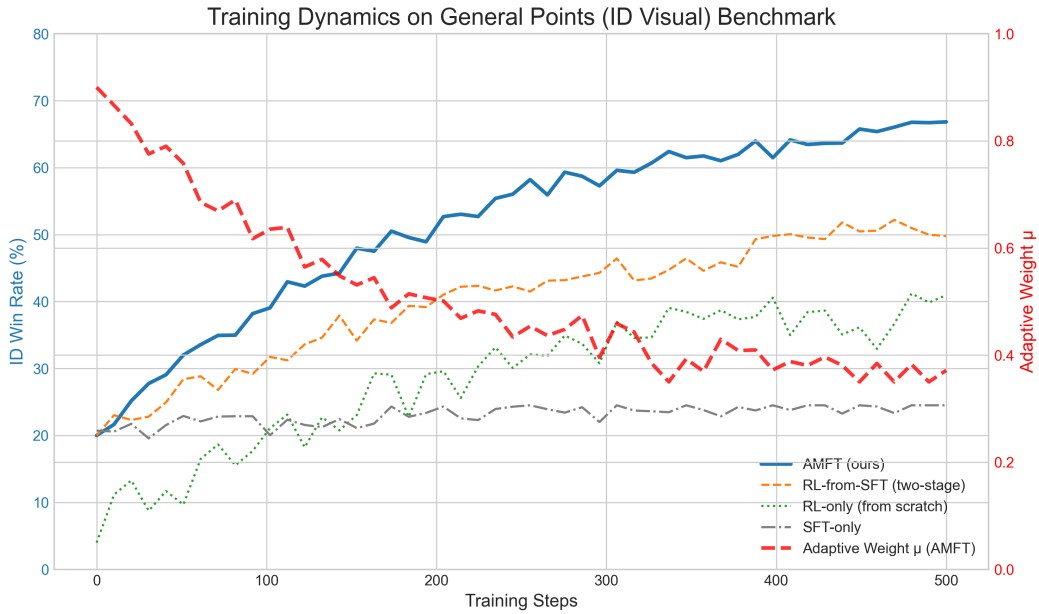

Figure 6: Learning dynamics on the **General Points** benchmark. The left y-axis shows the ID Win Rate (%), while the right y-axis shows the adaptive weight $\mu$. AMFT (solid blue) demonstrates a superior and more stable learning trajectory by dynamically adjusting $\mu$ (red dash-dotted) to learn an optimal curriculum, starting with SFT-dominance (high $\mu$) and smoothly transitioning to RL-dominance (low $\mu$).

**V-IRL (Vision-Language Navigation).** A similar, compelling pattern emerges in the V-IRL navigation task, a domain characterized by sequential decision-making, spatial reasoning, and the critical need to ground textual instructions in visual observations. As shown in Figure 7, AMFT once again achieves the highest final success rate with the most stable learning trajectory, confirming its effectiveness in a distinct reasoning domain.

**AMFT's Adaptive Strategy in a High-Baseline Scenario.** In this task, the **SFT-only** baseline performs strongly out of the gate, as the highly structured nature of navigation instructions (e.g., "turn left," "walk forward") is well-suited to imitation learning. However, its performance quickly plateaus, as it struggles to adapt to novel visual scenes or slightly ambiguous instructions not perfectly

represented in the training data. AMFT, by contrast, leverages this strong initial performance. Its controller begins with a high SFT weight ($\mu$) to efficiently absorb this foundational knowledge, matching the initial SFT trajectory.

Then, mirroring its behavior on General Points, the meta-controller autonomously and gradually reduces $\mu$, increasing the influence of the explicit RL reward. This transition is critical: it allows the model to move beyond imitating static paths and start learning a true, generalizable navigation policy. Through RL-driven exploration, the model refines its ability to ground textual commands (e.g., "The Dutch on your right") to specific visual landmarks in dynamic environments, correcting its own errors and ultimately pushing its performance beyond the imitation-based ceiling of the SFT and SFT→RL baselines. The trajectory of $\mu$ is again a clear testament to the controller's ability to learn an effective curriculum, moving from a phase of imitation to one of exploration and refinement.

**Contrasting with Baselines.** The performance of the baselines reinforces the narrative. While **RL-only** eventually learns a decent policy, its initial exploration is far less efficient than simply learning from the SFT data first. The **SFT→RL** pipeline is again a strong contender but its rigid, two-stage nature proves less effective than AMFT's dynamic balancing act. The smooth and principled transition orchestrated by AMFT's meta-controller avoids the potential instabilities of a hard switch and leads to a more robust and higher-performing final agent.

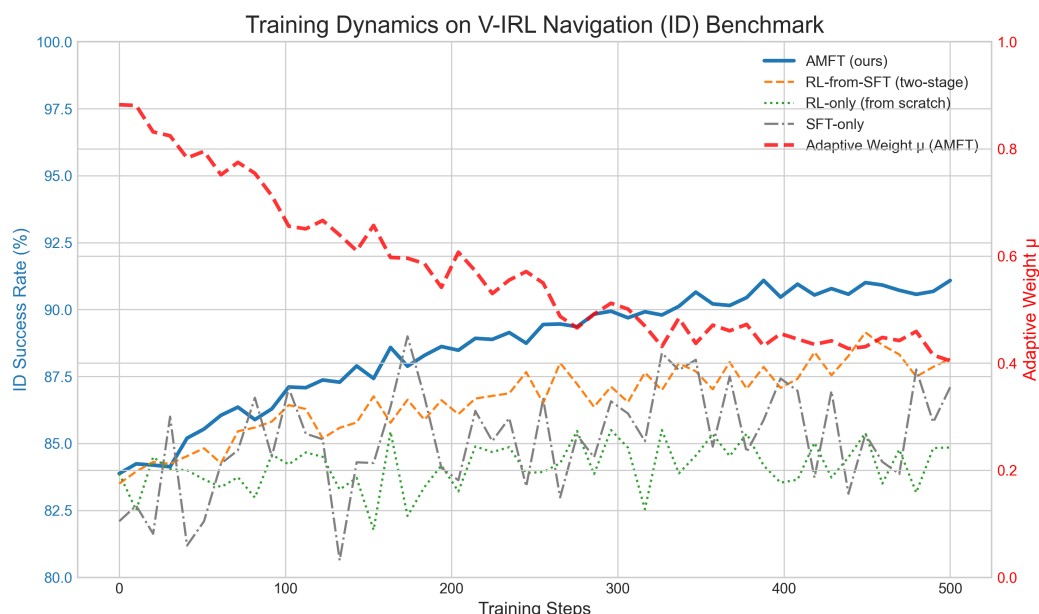

Figure 7: Learning dynamics on the **V-IRL Navigation** benchmark. The left y-axis shows the ID Success Rate (%), and the right y-axis shows the adaptive weight $\mu$. AMFT's learned curriculum, visualized by the red dash-dotted $\mu$ curve, enables it to build upon a strong SFT foundation and leverage RL to achieve the highest final performance, surpassing all baselines.

Collectively, these visualizations across two distinct multi-modal domains provide strong evidence for the general effectiveness and robustness of the AMFT framework. The meta-controller consistently learns a sensible curriculum—starting with imitation to build a stable foundation and progressively shifting to exploration to achieve superior performance—without any task-specific manual tuning. This demonstrates that our forward-looking, meta-learning approach provides a more principled and effective solution for balancing the SFT-RL trade-off than the reactive heuristics used in prior work.

### E.2 QUALITATIVE ANALYSIS: CASE STUDIES

While quantitative metrics provide a high-level summary of performance, a qualitative analysis of model outputs offers invaluable insight into the *behavioral differences* instilled by each fine-tuning

paradigm. This section presents case studies from our three reasoning domains to provide concrete examples of the phenomena discussed in the main paper, such as SFT's brittleness, RL's instability, and AMFT's balanced and effective reasoning.

**Case Study 1: Mathematical Reasoning.** This case study uses a moderately complex algebra problem. The goal is to assess whether the model has learned the underlying algebraic principles or has simply memorized solution templates.

---

**Problem Statement :**
Let $f(x) = x^2 + 6x + 7$. Find the vertex of the parabola. Your answer should be in the form $(h, k)$.

**Correct Reasoning and Answer:**
The vertex of a parabola $f(x) = ax^2 + bx + c$ is at $x = -b/(2a)$. Here, $a = 1, b = 6$. So, $h = -6/(2 \cdot 1) = -3$. To find $k$, we evaluate $f(h)$: $k = f(-3) = (-3)^2 + 6(-3) + 7 = 9 - 18 + 7 = -2$. The vertex is **(-3, -2)**.

---

Figure 8: Problem statement for the mathematical reasoning case study.

Table 8: Model outputs for the mathematical reasoning case study.

| SFT-only Output | RL-only (from scratch) Output | AMFT (ours) Output |
|---|---|---|
| To find the vertex, we complete the square. The function is $f(x) = x^2 + 6x + 7$. We rewrite this as $f(x) = (x^2 + 6x + 9) - 9 + 7$, which simplifies to $f(x) = (x + 3)^2 + 2$. From the vertex form $f(x) = a(x - h)^2 + k$, the vertex is $(h, k)$. So the vertex is (3, 2). | The vertex is the point where... vertex is the... point $a = 1, b = 6$, So $h = -3$. Then calculate x... x... f(x)... The vertex is (-3, 7). | To find the vertex $(h, k)$ of $f(x) = ax^2 + bx + c$, we can use $h = -b/(2a)$. In this case, $a = 1$ and $b = 6$. So $h = -3$. $k = f(-3) = (-3)^2 + 6(-3) + 7 = 9 - 18 + 7 = -2$. Therefore, the vertex of the parabola is (-3, -2). |
| *Analysis: Exhibits **brittle memorization**. The model correctly recalls the "completing the square" method. However, it makes a critical sign error when mapping $(x + 3)^2$ to the vertex form $(x - h)^2$. This demonstrates pattern imitation without a deep understanding of the underlying principles.* | *Analysis: Shows clear signs of **policy collapse**. The model fails to produce a coherent reasoning chain, getting stuck in repetitive and incomplete phrases.* | *Analysis: Demonstrates **robust and principled reasoning**. The model correctly identifies and applies the standard vertex formula, a more direct and less error-prone method. Each step is logically sound and mathematically correct.* |

**Case Study 2: Visual Arithmetic Reasoning (General Points).** This case study assesses the model's ability to ground its reasoning in visual input. The task requires both correct card recognition and subsequent arithmetic calculation.

---

**Problem Statement (General Points):**

You are observing these four cards in the image. Note that 'J', 'Q', and 'K' count as '10'. Your goal is to output a formula that evaluates to 24.
**Correct Reasoning and Answer:** The cards are Queen of Spades (10), 8 of Hearts, 3 of Clubs, and 2 of Diamonds. The numbers are 10, 8, 3, 2. The correct solution is **10*3-8+2=24**.

---

Figure 9: Problem statement for the visual arithmetic reasoning case study.

Table 9: Model outputs for the visual arithmetic reasoning case study.

| SFT-only Output | RL-only (from scratch) Output | AMFT (ours) Output |
|---|---|---|
| `{"cards": ["Q", "8", "3", "2"], "number": [10, 8, 3, 2], "formula": "(10 + 2) * (8 / 4)"}` | `{"cards": ["Q", "8", "K", "2"], "number": [10, 8, 10, 2], "formula": "10+8+10+2=30"}` | `{"cards": ["Q", "8", "3", "2"], "number": [10, 8, 3, 2], "formula": "10*3-8+2=24"}` |
| *Analysis: Exhibits **pseudo-thinking** and poor grounding. The model correctly identifies the cards and their values. However, it hallucinates the number '4' in its formula.* | *Analysis: Demonstrates both **poor visual perception** and **policy collapse**. The model incorrectly identifies the '3 of Clubs' as a 'King'. It then proceeds to perform a simple, incorrect addition.* | *Analysis: Shows **strong visual grounding and effective reasoning**. The model correctly identifies all four cards. It then successfully explores the solution space to find a valid, non-trivial formula that equals 24.* |

**Case Study 3: Vision-Language Navigation (V-IRL):**
**Instruction History:**
1. First, turn left to face east.
2. Move forward until you reach the next intersection where Hotel 32One is on your right behind.
3. Turn right to face north.
... (remaining steps)

**Action History:** A_1: turn_direction(east), A_2: forward(), A_3: forward()

**Current State:** The agent is at state 'O_4', observing an intersection via the 2x2 grid of street views.

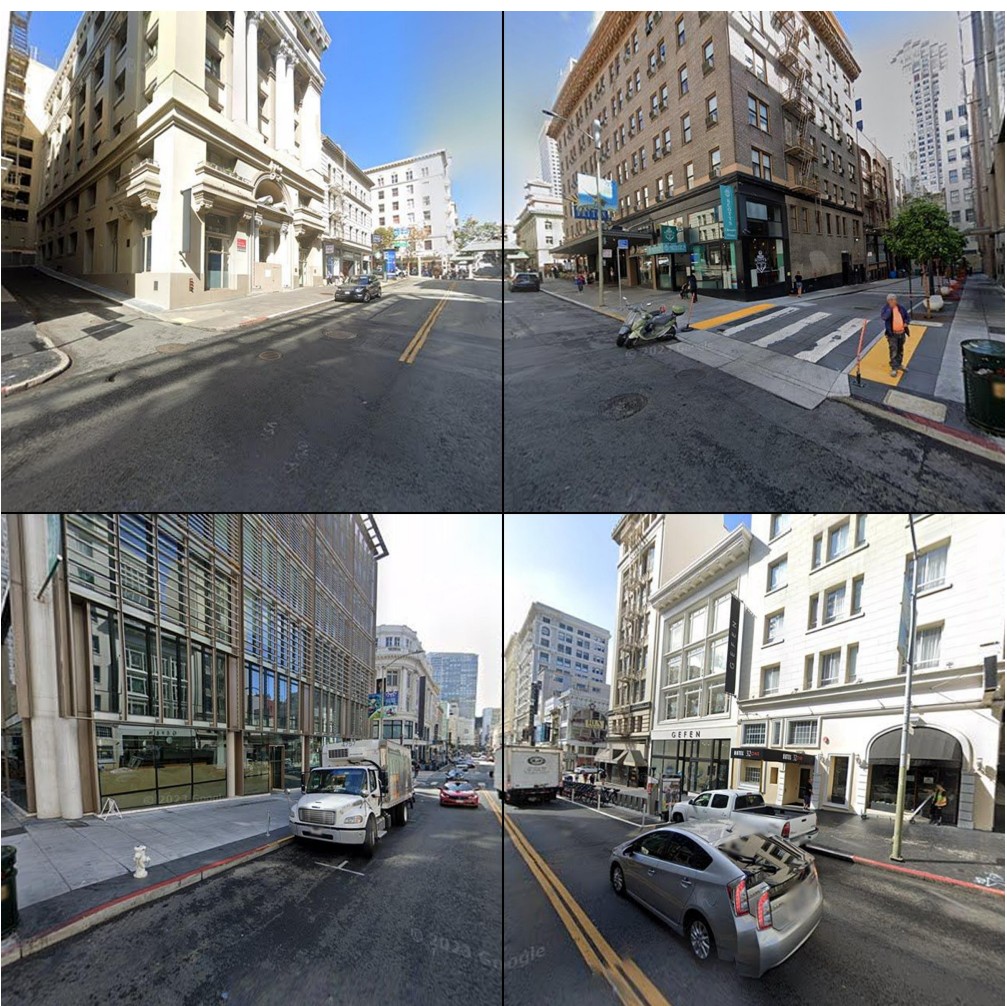

*The agent is facing east after the last turn, so "right behind" corresponds to the "back" view relative to its initial orientation, but from the current intersection, the key is simply to identify the landmark.*

**Correct Reasoning and Answer:**
The agent has completed instruction 2 by reaching the intersection with "Hotel 32One" visible. The next step is to execute instruction 3. **Action: turn_direction(north)**.

Figure 10: Problem statement for the revised vision-language navigation case study, based on the provided high-resolution visual grid.

Table 10: Model outputs for the revised vision-language navigation case study.

| SFT-only Output | RL-only (from scratch) Output | AMFT (ours) Output |
|---|---|---|
| `{"current observation": "I have reached the intersection and can see Hotel 32One in the right-side view.", "current instruction": "Move forward until you reach the next intersection where Hotel 32One is on your right behind.", "action": "forward()"}` | `{"current observation": "I see an intersection with several buildings.", "current instruction": "Turn right to face north.", "action": "stop()"}` | `{"current observation": "I have arrived at the intersection. I can confirm the presence of 'Hotel 32One' in the right-side view of the visual grid. This fulfills the condition of the current instruction.", "current instruction": "Turn right to face north.", "action": "turn_direction(north)"}` |
| *Analysis: Exhibits a critical failure in **state tracking and logical transition**. The model correctly performs visual perception, identifying "Hotel 32One". However, it becomes fixated on the current instruction and incorrectly repeats the associated action ('forward()'), failing to recognize that the terminal condition of that instruction has been met. This demonstrates a classic SFT pitfall: it has memorized the association between an instruction's text and an action, but has not learned the underlying state-machine logic required for multi-step plan execution.* | *Analysis: Demonstrates a severe lack of **visual grounding and policy coherence**. The model provides a generic, ungrounded observation ("several buildings") and fails to identify the key landmark. Although it happens to guess the next instruction textually, its chosen action ('stop()') is completely unrelated and nonsensical. This exemplifies policy collapse, where the absence of a stable, format-following policy from SFT leads to an inability to connect perception, planning, and action, resulting in random or useless behavior.* | *Analysis: Displays **accurate state tracking, robust visual grounding, and correct logical execution**. The model first grounds its action in the visual evidence, explicitly confirming "Hotel 32One". Crucially, it then correctly deduces that this observation satisfies the condition of instruction 2. This triggers a successful state transition, causing it to identify instruction 3 as the new goal and execute the correct corresponding action ('turn_direction(north)'). This showcases AMFT's ability to build a sophisticated and reliable decision-making policy that seamlessly integrates perception, reasoning, and planning.* |

## F    FURTHER STUDIES ON AMFT CONTROLLER

This section dissects the components of the AMFT adaptive weight controller to demonstrate that its specific design choices are crucial, well-justified, and robust. We conduct a series of studies investigating the sensitivity of the model's performance to the controller's key hyperparameters, including its learning rates, the frequency of meta-updates, and the target entropy setting.

### F.1    SENSITIVITY TO CONTROLLER HYPERPARAMETERS ($\eta_\mu, \eta_H$)

The AMFT controller's behavior is governed by two primary learning rates: the meta-gradient learning rate, $\eta_\mu$, which controls the influence of the long-term, forward-looking validation signal, and the entropy heuristic learning rate, $\eta_H$, which dictates the strength of the short-term, reactive stability signal. This study investigates how the final model performance varies across different settings of these two parameters. The goal is to demonstrate that while the performance is sensitive to these values, there exists a reasonably wide range where AMFT performs well, indicating that the method is robust and not prohibitively difficult to tune.

**Experimental Design.** We conducted a grid search over a range of values for $\eta_\mu$ and $\eta_H$, centered around the optimal configuration used in our main experiments ($\eta_\mu = 1 \times 10^{-4}, \eta_H = 5 \times 10^{-4}$). For each pair of hyperparameters, we ran a full 500-step training process on the mathematical reasoning task and evaluated the final checkpoint. Performance is reported as the average accuracy across the five in-distribution mathematical reasoning benchmarks (AIME24, AMC, MATH500, Minerva, and OlympiadBench).

**Results and Analysis.** The results of our sensitivity analysis are presented in Table 11. The data reveals several key insights into the controller's dynamics:

Table 11: Ablation study on the AMFT controller learning rates $\eta_\mu$ and $\eta_H$. Performance is reported as the average accuracy (%) on the five in-distribution mathematical reasoning benchmarks. The configuration used in the main paper is highlighted in bold.

| | | Meta-Gradient Learning Rate ($\eta_\mu$) | | |
| --- | --- | --- | --- | --- |
| | | $5 \times 10^{-5}$ | $1 \times 10^{-4}$ | $2 \times 10^{-4}$ |
| **Entropy** | $1 \times 10^{-4}$ | 60.1 | 59.8 | 58.7 |
| **Heuristic** | $5 \times 10^{-4}$ | 60.7 | **61.3** | 60.2 |
| **Learning Rate** | $1 \times 10^{-3}$ | 60.5 | 61.0 | 59.5 |

- **Existence of an Optimal Region:** There is a clear performance peak at $(\eta_\mu, \eta_H) = (1 \times 10^{-4}, 5 \times 10^{-4})$, which validates the hyperparameter choice for our main experiments. Importantly, the performance degrades gracefully around this peak rather than collapsing, with several neighboring configurations achieving strong results (e.g., $> 60.0\%$ accuracy). This demonstrates the robustness of the AMFT framework.

- **Impact of Meta-Gradient Rate ($\eta_\mu$):** The influence of the long-term signal is critical. When $\eta_\mu$ is too low ($5 \times 10^{-5}$), the controller adapts too slowly, failing to fully capitalize on the forward-looking signal to escape the suboptimal regions that heuristic-only methods might settle in. When $\eta_\mu$ is too high ($2 \times 10^{-4}$), the meta-updates to $\mu$ become too aggressive and potentially noisy, causing instability in the learned curriculum and slightly degrading final performance.

- **Impact of Entropy Heuristic Rate ($\eta_H$):** The short-term stabilizer is equally important. When $\eta_H$ is too low ($1 \times 10^{-4}$), the controller cannot react swiftly enough to policy entropy fluctuations. This makes the training less stable, especially if the meta-gradient is also high. Conversely, when $\eta_H$ is too high ($1 \times 10^{-3}$), the controller becomes overly reactive to transient entropy changes. This excessive regulation can dampen the long-term signal from the meta-gradient, preventing the policy from engaging in the necessary exploration, thus leading to a slightly suboptimal outcome.

In conclusion, this ablation study confirms that the performance of AMFT is dependent on a synergistic balance between its long-term, meta-learning objective and its short-term, stability-ensuring heuristic. The results show that while careful tuning is beneficial, the method is not overly sensitive to minor variations in its controller's learning rates, making it a robust and practical approach for fine-tuning LLM reasoners.

F.2 IMPACT OF META-UPDATE FREQUENCY ($K$)

The meta-gradient controller's update frequency, denoted by the hyperparameter $K$, is a critical factor that balances the controller's responsiveness against the computational cost of training. The meta-gradient $\nabla_\mu U(\theta_t)$ provides a principled, forward-looking signal, but its computation requires additional forward and backward passes on a validation set, making it more expensive than the main policy update. This ablation study investigates the impact of varying $K$ (the number of training steps between each meta-gradient update) on final model performance and overall training efficiency.

**Experimental Design.** We trained several AMFT models from the same SFT warm-up checkpoint, varying only the meta-update frequency $K$. We tested values ranging from very frequent updates ($K = 5$) to very infrequent updates ($K = 100$). All other hyperparameters, including the controller's

learning rates, were held constant at their optimal values as determined in Appendix F.1. Performance was measured by the average accuracy on the five in-distribution mathematical reasoning benchmarks after 500 training steps. We also report the total number of meta-gradient computations performed during training as a proxy for the additional computational overhead.

**Results and Analysis.** The results, presented in Table 12, clearly illustrate the trade-off between controller responsiveness and training efficiency. The data reveals three distinct operational regimes for the meta-controller based on the update frequency.

Table 12: Ablation study on the meta-update frequency $K$. Performance is the average accuracy (%) on in-distribution math benchmarks. The configuration used in the main paper is highlighted.

| Update Freq. ($K$) | Avg. Acc. (%) | Total Updates | Analysis and Observation |
|---|---|---|---|
| 5 | 60.9 | 100 | **High-Frequency:** Performance is slightly hindered by noisy, single-batch meta-gradients. |
| 10 | 61.1 | 50 | **Near-Optimal:** Strong performance with halved computational cost compared to K=5. |
| **20** | **61.3** | **25** | **Optimal Trade-off:** Effectively balancing controller responsiveness with computational and gradient stability. |
| 50 | 60.2 | 10 | **Low-Frequency:** The learned curriculum starts to lag behind the policy's needs. |
| 100 | 59.3 | 5 | **Very Low-Frequency:** The controller updates too infrequently, resulting in ineffective curriculum. |

- **High-Frequency Updates** ($K < 20$)**:** When the meta-gradient is computed very frequently (e.g., every 5 or 10 steps), the controller is highly responsive. However, this comes at a significant computational cost, as indicated by the high number of total meta-updates. Furthermore, these frequent updates can introduce instability, as each meta-gradient is estimated from a single, potentially noisy validation batch. This can cause the adaptive weight $\mu$ to oscillate unnecessarily, slightly hindering the model from settling into its optimal learning trajectory and resulting in a final performance just below the peak.

- **Optimal Trade-off Region** ($K \approx 20$)**:** The best performance (**61.3%**) is achieved at $K = 20$, the value used in our main experiments. At this frequency, the meta-updates are frequent enough to steer the training curriculum effectively, allowing $\mu$ to adapt to the policy's evolving needs in a timely manner. The interval is also long enough to average out some of the noise from single-batch gradient estimates and to significantly reduce computational overhead. This setting represents an empirically validated "sweet spot" that maximizes performance while maintaining practical training efficiency.

- **Low-Frequency Updates** ($K > 20$)**:** As the update frequency decreases ($K = 50$ or $K = 100$), the controller becomes progressively less responsive. The adaptive weight $\mu$ is updated too infrequently to keep pace with the policy's rapid learning in the inner loop. The learned curriculum becomes "stale," failing to provide the right balance of SFT and RL when the model needs it. This leads to a graceful but clear degradation in final performance. In the limit of $K \to \infty$, the training would be equivalent to using a fixed, manually-tuned $\mu$, forfeiting the benefits of dynamic, forward-looking adaptation.

In conclusion, this study validates our choice of $K = 20$ as a principled and effective compromise. It demonstrates that while the meta-learning component of AMFT is a vital driver of performance, its benefits can be realized without incurring the prohibitive computational costs associated with overly frequent updates.

