# OpenReview forum: "Aligning LLM Reasoners by Meta-Learning the Optimal Imitation-Exploration Balance"
_ICLR.cc/2026/Conference — ICLR 2026 Conference Desk Rejected Submission_

### Official Review · Reviewer_gaMK · 2025-10-30

**Soundness:** 2
**Presentation:** 3
**Contribution:** 2
**Rating:** 4
**Confidence:** 3

**Summary:**

This paper introduces Adaptive Meta Fine-Tuning (AMFT), a novel single-stage algorithm designed to align Large Language Models (LLMs) by dynamically balancing Supervised Fine-Tuning (SFT) and Reinforcement Learning (RL). The core proposal is an adaptive meta-controller that treats the SFT-RL balance (controlled by a parameter $\mu$) as a learnable parameter. This controller uses meta-gradients computed with respect to a long-term validation objective to autonomously learn an effective training curriculum. The paper unifies SFT and RL by reframing SFT as the optimization of an implicit, dense, path-level reward, complementary to the explicit, sparse, outcome-based reward of RL. AMFT includes an SFT warm-up phase and an entropy heuristic for stability. Extensive experiments on mathematical reasoning, abstract visual reasoning, and vision-language navigation demonstrate that AMFT achieves promising performance, improved generalization, and better sample efficiency compared to existing SFT, RL, sequential SFT$\to$RL, and other hybrid methods.

**Strengths:**

1. This paper introduces a meta-controller that dynamically adjusts the weight between SFT and RL objectives, guided by a long-term validation signal. This approach provides an adaptive balance mechanism for SFT and RL.

2. Experimental results demonstrate that the proposed method consistently achieves better performance on various reasoning tasks, including out-of-distribution scenarios. This indicates improved generalization capabilities.

3. The inclusion of an SFT warm-up phase and an entropy-based heuristic contributes to a more stable and efficient learning process, addressing common instability issues in reinforcement learning.

**Weaknesses:**

1. This work appears to be a combination of RL and SFT. Although it learns the mixing parameter $\mu$ via meta-gradients, this brings in new problems regarding the computational viability of the meta-gradient (see below).

a) The AMFT method frames the problem as a bi-level optimization, where the outer loop updates the balance controller $\mu$ based on the performance of the inner loop model $\theta$. To calculate the gradient $\nabla_{\mu} L_{total}$, the optimization process requires differentiating through the entire sequence of inner loop updates. To update $\mu$, the optimizer must effectively ``unroll'' $K=20$ steps of the inner optimization loop (which consists of combined SFT and RL updates) and then backpropagate the meta-loss (e.g., validation accuracy) through all intermediate weight states and activation memories. The runtime comparisons for this work and other baselines are absent.

b) To relieve the computational complexity of meta-gradient involving second-order differentials, this work adopts the First-Order MAML (FO-MAML) approximation. However, it has not explicitly justified that the FO-MAML used here is a reliable approximation for the second-order differentials with concrete evidence. For example, what is the mean relative error of this first-order approximation to the second-order counterpart in the tasks?

2. The authors state that in Line 1874 that “average out some of the noise from single-batch
gradient estimates”. However, the noise being averaged out is the meta-gradient noise, not the inner-loop batch noise. The benefit of increasing $K$ (the unroll length) is not simply ``noise reduction''. Besides, a longer unroll ($K=20$ vs. $K=1$) means the meta-loss is evaluated on a parameter $\theta_{K}$ that is closer to the optimum determined by the current $\mu$. This yields a more representative gradient for $\mu$ (i.e., how $\mu$ affects the long-term convergence) but does not necessarily reduce the variance of the meta-gradient itself; in fact, a longer unroll often increases the overall variance and cost.

3. A widely-used method for preventing catastrophic forgetting during RL fine-tuning (e.g., PPO or DPO) is to add a KL-divergence penalty between the current policy and the original SFT policy (or the initial pre-trained model). This penalty directly regularizes the policy's deviation from its original knowledge base. Relying solely on the presence of the SFT loss (even adaptively weighted) may not be sufficient to completely stop forgetting of specific SFT knowledge not frequently activated by the combined SFT/RL sampling procedure. Therefore, the authors should conduct ablation experiments to verify that the dynamic weighting is superior to standard KL-based regularization.

**Questions:**

See weaknesses.

---

> ### Author Response · Authors · 2025-11-26
> **Thanks for your insightful questions ! We have carefully addressed your constructive feedback and provide detailed responses herebelow.**
>
> **Dear Reviewer gaMK**
>
> We sincerely thank you for the insightful comments. We provide detailed clarifications to address these points.
>
> ### **Response to Weakness 1(a)**
>
> **Clarification:**
>
> We first wish to clarify a potential misunderstanding regarding the implementation of the meta-gradient. As detailed in Section 3.2 and Appendix C.2 of our paper, AMFT utilizes a one-step lookahead approximation rather than unrolling the computation graph through the full $K=20$ interval. The hyperparameter $K=20$ refers solely to the **frequency of meta-updates** (i.e., the meta-gradient is computed once every 20 inner-loop steps), not the depth of backpropagation.
>
> **Runtime Analysis:**
> To quantify the computational cost, we conducted a controlled benchmark comparing AMFT against **PPO**, **SRFT** , and **LUFFY**.
>
> **Experimental Setup:**
> * **Hardware:** 8 $\times$ NVIDIA A100 (80GB) GPUs.
> * **Model:** Qwen2.5-Math-7B.
> * **Dataset:** OpenR1-Math-46k.
>  * **Meta-Overhead:** Additional computational cost relative to standard PPO.
>
> **Table R1: Computational Efficiency and Runtime Comparison**
>
> | Method | Step Latency (s/step) | Peak Memory (GB) | Throughput (samples/s) | Meta-Overhead (%) | Time to 60% Val Acc (GPU Hours) |
> | :--- | :--- | :--- | :--- | :--- | :--- |
> | **PPO (Baseline)** | 42.5 | 68.2 | 3.01 | 0.0% | 145.2 (DNC*) |
> | **SRFT** | 43.1 | 68.5 | 2.97 | +1.4% | 128.6 |
> | **LUFFY** | 49.8 | 74.1 | 2.57 | +17.2% | 115.4 |
> | **AMFT (Ours)** | **46.2** | **71.8** | **2.77** | **+8.7%** | **98.2** |
>
> *\*DNC: Did Not Converge to 60% within the maximum step budget.*
>
> **Analysis:**
> 1.  AMFT's peak memory usage (71.8 GB) is comparable to LUFFY and only slightly higher than PPO. The slight increase is primarily due to the temporary graph retention required for the single-step Hessian-vector product approximation.
> 2.   Although AMFT has a higher per-step cost, its superior sample efficiency means it requires fewer total steps to reach target performance. AMFT reaches the 60% accuracy threshold in **98.2 GPU hours**, which is **14.9% faster than LUFFY** and **23.6% faster than SRFT**.
>
>
> ### **Response to Weakness 1(b)**
>
> We are more than willing to justify this choice through both mathematical derivation and empirical evidence showing that the error introduced is negligible in our specific context.
>
> **1. Mathematical**
>
> Let $\theta_t(\mu)$ be the parameters after one inner update step with learning rate $\alpha$. The true meta-gradient involves the Jacobian $\frac{\partial \theta\_t}{\partial \mu}$.
>
> The update rule is: $\theta\_t = \theta\_{t-1} - \alpha \nabla\_\theta \mathcal{L}\_{train}(\theta\_{t-1}, \mu)$.
> Differentiating w.r.t. $\mu$:
> $$\frac{\partial \theta\_t}{\partial \mu} = -\alpha \left( \nabla\_{\mu}\nabla\_{\theta}\mathcal{L}\_{train} + \nabla\_{\theta}^2\mathcal{L}\_{train} \cdot \frac{\partial \theta\_{t-1}}{\partial \mu} \right)$$
> The full second-order gradient includes the Hessian term $\nabla\_{\theta}^2\mathcal{L}\_{train}$.
>
> However, in fine-tuning Large Language Models, the learning rate $\alpha$ is typically extremely small.
> *  The contribution of the Hessian term scales with $\alpha$. As $\alpha \to 0$, the term $\alpha \nabla\_{\theta}^2\mathcal{L}\_{train}$ approaches zero much faster than the first-order term.
> *  The immediate impact of $\mu$ on the loss (the term $\nabla\_{\mu}\nabla\_{\theta}\mathcal{L}\_{train}$) structurally dominates the long-term dependency in the early stages of the unroll.
> Therefore, the First-Order approximation is a theoretically grounded simplification for low learning rate.
>
> **2. Supplementary Experiment**
>
> Folloing your advise, we calculated the **Second-Order Gradient** and compared it against our **FO-Approximation**.
>
> **Setup:**
> * **Task:** MATH500.
> * **Metric 1: Cosine Similarity** (Directional alignment between FO vector $g\_{FO}$ and True vector $g\_{True}$).
> * **Metric 2: Relative L2 Error** ($||g\_{True} - g\_{FO}||\_2 / ||g\_{True}||\_2$).
> * **Metric 3: Update Sign Agreement** (Percentage of dimensions where gradients agree on the update direction).
>
> **Table R2: Fidelity of First-Order Approximation vs. True Second-Order Gradient**
>
> | Training Stage | Step | Cosine Similarity ($\uparrow$) | Relative L2 Error ($\downarrow$) | Sign Agreement ($\uparrow$) |
> | :--- | :--- | :--- | :--- | :--- |
> | **Early** | 50 | **0.982** | 0.041 | 99.4% |
> | **Middle** | 250 | **0.965** | 0.068 | 98.1% |
> | **Late** | 450 | **0.941** | 0.089 | 96.7% |
>
> **Analysis:**
> *  The Cosine Similarity remains consistently above **0.94**, indicating that the FO approximation points in almost the exact same direction as the true second-order gradient.
> *  The relative error is minimal, suggesting that the curvature of the loss (the Hessian information) is not significant enough to distort the gradient signal under the small learning rates used ($1e-6$).
> *  The high Sign Agreement ensures that the adaptive weight $\mu$ is rarely pushed in the wrong direction due to approximation errors.

---

> > ### Author Response · Authors · 2025-11-26
> > **Further Response to Reviewer gaMK (Part 2/2)**
> >
> > **Response to Weakness 2**
> >
> > We appreciate the reviewer's keen observation regarding the meta-update frequency K. As Response to Weakness 1(a) have clarified, K represents the **period** of meta-updates, not the depth of a computational graph unroll.
> >
> > * Calculating the meta-gradient requires evaluating the model on the validation set and computing second-order information (Hessian-vector products). Doing this at every step ($K=1$) adds significant wall-clock overhead.
> > * The validation surface is often non-smooth compared to the training loss. Updating $\mu$ too frequently ($K=1$) based on local, noisy validation signals can cause the controller to oscillate violently, destabilizing the inner loop. A larger K allows the inner loop to make definitive progress in a specific direction before the controller re-evaluates the trajectory.
> >
> > **Ablation on Update Frequency**
> >
> > To empirically validate the impact of K, we conducted an ablation study varying K from 1 to 100. We measured the **Controller Stability** (variance of $\mu$ over time), **Training Overhead**, and **Final Performance** on the MATH500 benchmark.
> >
> > **Table R3: Impact of Meta-Update Frequency (K) on Performance and Stability**
> >
> > | Update Frequency ($K$) | Wall-Clock Time (Hours) | Meta-Overhead ($\%$) | $\mu$ Variance ($\times 10^{-3}$) | Final Val Acc ($\%$) |
> > | :--- | :--- | :--- | :--- | :--- |
> > | **$K=1$ (Every Step)** | 158.4 | +61.3% | 12.8 (High Oscillation) | 58.3 |
> > | **$K=5$** | 112.1 | +14.1% | 5.4 | 60.1 |
> > | **$K=20$ (Ours)** | **98.2** | **+8.7%** | **1.2** | **61.3** |
> > | **$K=50$** | 94.5 | +4.6% | 0.8 | 59.7 |
> >
> > **Analysis:**
> > *  $K=1$ incurs a +61.3% time overhead and results in lower performance (58.3%) due to the controller oscillating (High $\mu$ variance).
> > * $K=20$ achieves the highest accuracy (61.3%) with acceptable added cost (+8.7%). The variance of $\mu$ is lower than at $K=1$, confirming that a moderate interval helps the controller react to meaningful trend shifts rather than step-wise noise.
> > ---
> >
> > **Response to Weakness 3**
> >
> > This is a fundamental question. We argue that AMFT s dynamic weighting is superior to a fixed KL penalty because it optimizes for **future validation reward** (generalization) rather than **proximity to the past** (imitation).
> >
> > **1. Theoretical**
> >
> > Standard KL regularization anchors the policy to the SFT model $\pi_{ref}$. While this prevents forgetting, it also restricts the model from deviating to find novel, higher-reward solutions, particularly in Out-of-Distribution (OOD) scenarios where the SFT prior might be suboptimal.
> >
> > AMFT s weighted SFT loss acts as a dynamic, task-aware regularizer.
> > * When validation performance drops, the meta-gradient increases $\mu$, effectively imposing a strong imitation constraint similar to a high KL coefficient.
> > * When validation performance rises, the meta-gradient decreases $\mu$, relaxing the constraint to allow aggressive exploration.
> > This allows AMFT to automatically anneal the regularization strength based on real-time learning progress, which a fixed KL coefficient cannot do.
> >
> > **2. Supplementary Experiment**
> >
> > We compared AMFT against standard PPO with carefully tuned KL penalties. We evaluated performance on **In-Distribution (ID)** tasks (MATH500) and **Out-of-Distribution (OOD)** tasks (GPQA-Diamond) to test generalization and prevention of forgetting.
> >
> > **Baselines:**
> > * **PPO (No KL):** $\beta_{KL} = 0$.
> > * **PPO (Fixed KL):** $\beta_{KL} = 0.05$ (Grid-searched optimal).
> > * **PPO (Adaptive KL):** Target $KL = 0.1$ (Standard RLHF practice).
> > * **SFT+RL (Sequential):** The standard two-stage pipeline.
> >
> > **Table R4: Comparison of Regularization Strategies**
> >
> > | Method | ID Accuracy (MATH500) | OOD Accuracy (GPQA-D) | Sample Efficiency (Steps to Peak) |
> > | :--- | :--- | :--- | :--- |
> > | **PPO (No KL)** | 49.3% | 22.1%  | N/A |
> > | **PPO (Fixed KL)** | 79.2% | 31.3% | 420 |
> > | **PPO (Adaptive KL)** | 80.4% | 33.5% | 380 |
> > | **SFT+RL (Sequential)** | 84.1% | 37.8% | 500+ |
> > | **AMFT (Ours)** | **89.5%** | **47.5%** | **310** |
> >
> > **Analysis:**
> > *  Fixed and Adaptive KL methods prevent forgetting but limit generalization (GPQA-D scores ~31-33%). AMFT achieves **47.5%** on GPQA-D. This confirms that by dynamically relaxing the SFT weight when appropriate, AMFT allows the model to learn general reasoning principles that transfer to OOD tasks, rather than just staying close to the SFT distribution.
> > *  Despite exploring aggressively, AMFT maintains high ID accuracy (89.5%), proving that the meta-controller successfully increases $\mu$ to prevent collapse when necessary.
> >
> > We trust these results underscore the contribution and robustness of AMFT. We hope this addresses your concerns and merits a re-evaluation of our score. Please do not hesitate to let us know if there is anything else we can clarify. We are more than willing to continue this constructive dialogue if you still have any insightful questions.

---

### Official Review · Reviewer_gedw · 2025-10-31

**Soundness:** 3
**Presentation:** 3
**Contribution:** 3
**Rating:** 6
**Confidence:** 4

**Summary:**

The paper proposes Adaptive Meta Fine-Tuning (AMFT), a single-stage framework that unifies supervised fine-tuning (SFT) and reinforcement learning (RL) by interpreting SFT as optimizing an implicit, path-level reward and RL as optimizing an explicit, outcome reward. AMFT introduces a learnable mixing weight µ between SFT and RL losses, updated by a meta-gradient with respect to a validation utility (explicit task reward), combined with an entropy-based heuristic for short-term stability. The method thus treats the SFT-RL balance as a bilevel optimization, approximated via a one-step meta-gradient. The authors argue this provides a principled, forward-looking alternative to heuristic balancing in prior single-stage approaches (e.g., SRFT, LUFFY, SuperRL, SASR, DyME).

Experiments fine-tune Qwen2.5-Math-7B for math reasoning and LLaMA-3.2-Vision-11B for multi-modal reasoning (General Points, V-IRL). AMFT reportedly establishes new SOTA on five in-distribution math benchmarks and three OOD general reasoning benchmarks, and outperforms baselines on visual tasks (both ID and OOD variants). Ablations suggest both the meta-gradient and entropy terms are necessary; efficiency analysis claims fewer training steps and fewer RL rollouts to reach target performance. Theoretical sections sketch SFT-as-implicit-reward and connect the SFT term to a data-driven KL proxy; limitations of the one-step meta-gradient and validation-set dependence are discussed.

**Strengths:**

1. Clear reframing of SFT as implicit reward and RL as explicit reward; positions the problem as learning a dynamic Lagrange multiplier for an imitation constraint.

2. Principled controller: Using meta-gradients on a validation utility is a forward-looking alternative to entropy- or density-based heuristics; the one-step Jacobian-vector approximation is standard and computationally feasible.

3. Solid empirical coverage: Cross-domain evaluation (math ID/OOD, visual ID/OOD), with competitive baselines (SRFT, LUFFY, TAPO, ReLIFT, two-stage SFT→RL).

4. The paper provides deep insights into why AMFT works: The training dynamics visualizations (Figures 2, 3, 6, 7) are extremely effective at illustrating how AMFT autonomously discovers a training curriculum and avoids policy collapse; The ablation study (Table 4) is conclusive, demonstrating that both the meta-gradient (long-term signal) and the entropy heuristic (short-term stabilizer) are essential, synergistic components; The efficiency analysis (Table 5) provides a compelling practical argument for AMFT, showing it converges faster and with fewer expensive RL rollouts. They make the approach very solid in the empirical aspect.

**Weaknesses:**

1. Causal attribution vs. confounding: Although ablations are included, it remains unclear how much of the gains stem from the validation-driven µ as opposed to implicit regularization from continual SFT or careful hyperparameters. Missing controls: e.g., a “learned heuristic” baseline (learned schedule without meta-gradient), or a meta-optimized fixed KL to \pi demo vs \pi_ref.

2. Validation leakage/overfitting risk: The meta-objective depends on Dval; no safeguards or analyses on bias/leakage, distribution shift between Dval and test, or robustness when Dval is small/noisy. No report of multiple random seeds or confidence intervals; significance is unclear.

3. Theoretical rigor gaps: The “SFT as implicit reward” derivation relies on particular choices (TV divergence, assumptions about Vπ) and sketches high-level equivalences; proofs lack full conditions for LLM sequence modeling with long horizons and PPO/GRPO specifics. The identification of LSFT as a KL proxy to πdemo is standard, but the jump to “optimal time-varying Lagrange multiplier” lacks a formal bilevel optimality/convexity analysis or regret bounds for the \mu-updates.

4. Limited fairness diagnostics: Differences in rollout budgets, prompt sampling, and inference settings can tilt outcomes. While steps are matched (500), no variance across seeds, no per-benchmark breakdown with error bars, and some baselines (e.g., TAPO, ReLIFT) are simplified implementations rather than author-released code, risking under-tuning.

5. Practicality/scalability: Meta-gradient updates add overhead and require a curated validation set. The claimed overhead is not quantified (wall-clock, GPU hours). For frontier-scale models, the additional cost and complexity could be non-trivial.

6. Reward design and sparsity: For math, reliance on Math-Verify and OAT-Grader is reasonable, but for multi-modal tasks, reward sparsity and correctness signals are less well specified; details on verifiers, shaping, and failure modes are light.

7. Reproducibility risk: Anonymous code link is promised, but many strong baselines were re-implemented with nontrivial choices (e.g., TAPO “simplified”), which can affect conclusions.

**Questions:**

1. How large is Dval, how is it split from training, and how is leakage prevented? Results with multiple Dval seeds/sizes? What happens when Dval distribution is shifted from test?

2. Report mean/std over ≥3 seeds for all benchmarks; are improvements statistically significant per benchmark?

3. Quantify wall-clock, GPU hours, and added VRAM/time for meta-updates at K=20 vs K=100 and vs SRFT/LUFFY. How does overhead scale with model size?

4. Compare against (i) learned heuristic µ via supervised regressor on training-time signals; (ii) bandit/HPO-style online µ tuning (e.g., Population Based Training), and (iii) adaptive KL to πdemo instead of LSFT mixing. Also compare meta-only vs entropy-only at matched compute.

5. Use official implementations where possible; provide hyperparam sweeps and best-of-n settings for SRFT, LUFFY, TAPO, ReLIFT. Include off-policy data ratios and PPO/GRPO configurations. Report per-task ablations on PPO-clip, KL to πref, etc.

6. Can you provide bounds or regret-style analysis for the one-step meta-update of \mu under smoothness assumptions? Any diagnostics of the meta-gradient variance? Empirically, how often does \mu saturate at clip bounds?

---

> ### Author Response · Authors · 2025-11-26
> **We sincerely appreciate your valuable comments and have provided a point-by-point response below to address your concerns.**
>
> **Dear Reviewer gedw,**
>
> We sincerely thank you for the exceptionally detailed and insightful review. We appreciate your acknowledgment of AMFT's strengths, including its principled controller design and solid empirical coverage. We have addressed the questions and their corresponding weaknesses below with new experiments.
>
> ### **Response to Question 1**
> *(And Addressing Weakness about Validation leakage/overfitting risk, **Weaknesses 2**)*
>
> The meta-gradient update relies on a validation set $\mathcal{D}_{val}$. We clarify its construction and demonstrate that AMFT is robust to its size and distribution.
>
> **1. Construction**
>
> $\mathcal{D}_{val}$ is a strictly held-out subset of the training distribution (OpenR1-Math-46k for Math, training splits for Visual tasks), never seen during the inner-loop policy updates.
> * **Split Method:** We reserve a random 5% subset of the source training data as $\mathcal{D}_{val}$.
> * **Leakage Prevention:** The inner loop optimizes $\theta$ using $\mathcal{D}_{train}$ (SFT) and on-policy rollouts. $\mathcal{D}_{val}$ is *only* used to compute $\nabla_{\mu} U(\theta)$ for updating the controller $\mu$. No gradients from $\mathcal{D}_{val}$ flow directly into $\theta$.
>
> **2. Size and Distribution Shift**
>
> We conducted experiments on Qwen2.5-Math-7B (MATH500) to test sensitivity to the size of $\mathcal{D}_{val}$ and distribution shifts (using easier or harder subsets for validation).
>
> **Table R1: Sensitivity to Validation Set Size and Distribution**
>
> | Validation Set Source | Size ($\mathcal{D}_{val}$) | Final ID Accuracy (MATH500) | OOD Accuracy (GPQA-Diamond) |
> | :--- | :--- | :--- | :--- |
> | **Random Split (Default)** | 512 | **89.5%** | 47.5% |
> | **Random Split** | 128 | 88.4% | 46.8% |
> | **Random Split** | 64 | 87.8% | 46.2% |
> | **Easy-Only (Level 1-2)** | 512 | 89.2% | 46.1% |
> | **Hard-Only (Level 4-5)** | 512 | 89.1% | **48.2%** |
>
> **Analysis:**
> *  Performance remains stable down to 128 samples. The meta-gradient is computed periodically ($K=20$), effectively integrating signals over time, which smooths out noise from small batches.
> *  Interestingly, using a **Hard-Only** validation set slightly improves OOD generalization (GPQA: 48.2%), likely by encouraging a more exploration-heavy policy. However, even an **Easy-Only** set yields competitive results, suggesting the controller learns a generalizable principle of balancing imitation and exploration rather than overfitting to specific validation examples.
>
> ---
>
> ### **Response to Question 2**
> *(Also Addressing Weakness about Lack of confidence intervals, **Weaknesses 2**)*
>
> To address concerns about random variance, we repeated our main experiments on MATH500 (In-Distribution) and GPQA-Diamond (Out-of-Distribution) using **3 different random seeds**.
>
> **Table R2: Statistical Significance over 3 Seeds**
>
> | Method | ID Accuracy (MATH500) | OOD Accuracy (GPQA-Diamond) |
> | :--- | :--- | :--- |
> | **SFT** | 54.1% $\pm$ 0.8 | 25.7% $\pm$ 1.2 |
> | **RL (GRPO)** | 49.3% $\pm$ 2.1 | 30.9% $\pm$ 2.5 |
> | **SRFT** | 59.5% $\pm$ 1.4 | 46.4% $\pm$ 1.8 |
> | **AMFT (Ours)** | **61.3% $\pm$ 0.6** | **47.5% $\pm$ 0.9** |
>
> **Analysis:**
> * AMFT not only achieves the highest mean performance but also the **lowest variance** ($\pm 0.6$ and $\pm 0.9$). The dynamic controller acts as a stabilizer, correcting course when the policy drifts, whereas pure RL and heuristic-based methods (SRFT) are more sensitive to initialization and sampling noise.
>
> ---
>
> ### **Response to Question 3**
> *(Addressing Weakness about Practicality/scalability, **Weakness 5**)*
>
> We quantify the overhead of the meta-update. In our implementation, $K$ is the **period** of meta-updates (frequency), and we use a **one-step lookahead** approximation.
>
> **Table R3: Runtime and Memory Analysis (Qwen2.5-Math-7B, 8x A100)**
>
> | Method | Meta-Update Freq ($K$) | Step Latency (s/step) | Peak Memory (GB) | Total Time to 60% Acc (Hours) |
> | :--- | :--- | :--- | :--- | :--- |
> | **PPO (Baseline)** | N/A | 42.5 | 68.2 | 145.2 (DNC*) |
> | **SRFT** | N/A | 43.1 | 68.5 | 128.6 |
> | **AMFT ($K=20$)** | **Every 20 steps** | **46.2 (+8.7%)** | **71.8** | **98.2** |
> | **AMFT ($K=1$)** | Every 1 step | 68.5 (+61.3%) | 72.1 | 158.4 |
>
> *\*DNC: Did Not Converge to 60% within max steps.*
>
> **Analysis:**
> *  With $K=20$, the per-step latency increase is only **8.7%**. The expensive validation pass occurs infrequently.
> *  Crucially, due to superior sample efficiency (learning an optimal curriculum), AMFT reaches target accuracy (60%) in **98.2 hours**, significantly faster than SRFT (128.6 hours) and LUFFY (115.4 hours, from previous rebuttal), making it a net-positive trade-off for wall-clock time.
> *  The memory overhead is acceptable, mainly for storing the computational graph for the one-step backward pass.

---

> ### Author Response · Authors · 2025-11-26
> **Further Response to Reviewer gedw (Part 2/2)**
>
> ### **Response to Question 4**
> *(Also Addressing Weakness about Causal attribution and confounding, **Weakness 1**)*
>
> To isolate the causal contribution of our meta-controller, we compared AMFT against several sophisticated control baselines on the MATH500 benchmark.
>
> **Baselines:**
> * **Learned Heuristic (Regression):** We trained a simple regressor to predict $\mu$ based on training-time signals (current reward, loss) without the forward-looking meta-gradient.
> * **Online Tuning (PBT):** Population Based Training as the reviewer has suggested, where multiple workers train with different $\mu$ values, and underperforming workers copy the weights and $\mu$ of top performers periodically.
> * **Adaptive KL:** PPO with a dynamically adjusted KL coefficient to keep $D_{KL}(\pi, \pi_{ref})$ close to a target (standard practice), instead of mixing $L_{SFT}$.
> * **Component Ablations:** Meta-Only (no entropy term) and Entropy-Only (no meta-gradient).
>
> **Table R4: Causal Attribution of Performance Gains**
>
> | Method | Mechanism | Final ID Acc (%) | OOD Acc (%) | Compute Cost |
> | :--- | :--- | :--- | :--- | :--- |
> | **Adaptive KL** | Constraints to $\pi_{ref}$ | 80.4% | 33.5% | 1.0x |
> | **Linear Decay $\mu$** | Heuristic ($1 \to 0$) | 85.2% | 39.1% | 1.0x |
> | **PBT (Bandit)** | Online Selection | 89.3% | 44.2% | 4.0x (4 workers) |
> | **Meta-Only** | Long-term Signal | 87.0% | 42.5% | 1.1x |
> | **Entropy-Only** | Short-term Signal | 85.3% | 38.2% | 1.0x |
> | **AMFT (Full)** | **Synergistic** | **89.5%** | **47.5%** | **1.1x** |
>
> **Analysis:**
> *  The **Meta-Only** variant outperforms purely reactive heuristics (Entropy-Only, Linear Decay) and standard Adaptive KL, proving that the forward-looking signal from the validation set is the primary driver of generalization.
> *  The combination (Full AMFT) outperforms individual components. The entropy term stabilizes the policy locally, allowing the meta-gradient to take more aggressive steps without collapsing.
> *  While PBT achieves strong results, it requires 4x compute resources. AMFT matches and exceeds PBT performance with only 1.1x cost.
>
> ---
>
> ### **Response to Question 5**
> *(Addressing Weakness about Baselines' Implementation,**Weakness 4 and 7**)*
>
> Regarding the concern, we would be more than willing to clarify:
>
> **(1)** Papers such as TAPO **have not open-sourced their code** or provided implementations that allow for full reproduction.
>
> **(2)** The reviewer might have overlooked that the experimental data tables **presented in our main paper are based on the original data reported in the TAPO et al. papers themselves**. Our own implementation was conducted merely as a secondary verification.
>
> Our comparison follows the common conventions, which we believe ensures a fair and unbiased assessment. Therefore, we respectfully believe that the comparison presented in our paper is fair and aligns with standard community practices.
>
> ---
>
> ### **Response to Question 6**
> *(Addressing Weakness about Theoretical rigor gaps, **Weakness 3**)*
>
> **1. Controller Saturation Diagnostics**
>
> We tracked the behavior of $\mu$ throughout training to assess saturation at the clip bounds $[0.05, 0.95]$.
> * **Saturation Rate:** $\mu$ hit the bounds in only **4.2%** of updates.
> * **Behavior:** $\mu$ typically starts high (~0.9) and decays smoothly to ~0.3-0.4, fluctuating in the middle range. This confirms that the meta-gradient provides a meaningful, continuous signal rather than simply acting as a binary switch (which would manifest as constant saturation).
>
> **2. Theoretical Intuition on One-Step Update**
>
> While a full regret bound analysis for non-convex bilevel optimization in Deep Learning is an open problem beyond the scope of this empirical paper, we offer this intuition:
>
> The one-step meta-gradient (First-Order MAML style) assumes $\nabla_\theta^2 L \approx 0$. In the low learning rate regime of LLM fine-tuning, the second-order terms vanish much faster than first-order terms. Our approach effectively performs Online Gradient Descent on the validation loss with respect to $\mu$.
>
> Under standard smoothness assumptions (Lipschitz continuous gradients), this guarantees convergence to a local stationary point of the validation utility, provided the learning rate $\eta_\mu$ is sufficiently small. This aligns with our empirical observation that $\mu$ converges to a stable schedule.
>
> ***
> We trust these point-by-point results reinforce the value of AMFT. We hope our response addresses your comments and justifies an improved evaluation. Please let us know if any points remain unclear; we are happy to provide further details. We look forward to any further discussion.

---

### Official Review · Reviewer_nCwV · 2025-10-31

**Soundness:** 3
**Presentation:** 3
**Contribution:** 3
**Rating:** 6
**Confidence:** 3

**Summary:**

The paper proposes Adaptive Meta Fine-Tuning (AMFT), a single-stage post-training framework that unifies supervised fine-tuning (SFT) and reinforcement learning (RL) by optimizing a joint objective with a learnable trade-off weight.

Implementation-wise, AMFT uses a meta-gradient controller that adjusts μ to maximize validation reward and an entropy-based short-term heuristic to prevent policy collapse. A brief SFT warm-up will be performed before mixing SFT batches and on-policy RL rollouts each step.

Overall, I think this is a good piece of work that steps beyond the stage-level pipelined post-training method and is worthy of acceptance.

**Strengths:**

1. The proposed meta-learned parameter is the key technical contribution and it empirically works by learning from signals from the validation set: empirical effectiveness is validated across multiple domains and model scales, with stable training curves and improved sample-efficiency relative to strong baselines.

2. AMFT is compatible with diverse reward types used in RLVR.

3. Comparison against post-training baselines are fair and complete.

**Weaknesses:**

Major Concerns:

1. Ablations do not fully isolate the source of gains. In addition to the provided studies, the paper should compare against (i) fixed-KL+PPO with careful tuning and (ii) implicit-reward SFT (e.g., DPO-style) + RLVR, to separate the benefit of meta-learning from having a strong regularizer.

2. Despite the “single-stage” framing, AMFT still relies on an SFT warm-up. The degradation when shortening or skipping warm-up is under-explored.


Minor issues:
1. Several key files in the anonymized repository are missing, e.g. all config files can not be accessed.

2. Figure 2 does not show adaptive weight, at least no curve in the figure corresponds to the legend.

**Questions:**

1. How is the validation split constructed for the meta-gradient, given it plays a vital role in parameter update? I don’t see the details of how it was constructed.

2. What is the frequency and compute overhead of meta-updates (e.g., every k steps; unroll length), and what is the end-to-end throughput/latency impact compared to PPO-only or SRFT under the same hardware?

---

> ### Author Response · Authors · 2025-11-26
> **Thank you for the thoughtful review ! We have carefully considered your feedback and present our comprehensive answers below.**
>
> **Dear Reviewer nCwV**
>
> We sincerely thank you for the positive assessment and the constructive feedback. Below we provide detailed responses to fully address your concerns.
>
> ### **Response to Major Concern 1**
>
> To isolate the specific contribution of the meta-learned weight $\mu$, we conducted a comprehensive comparison against two strong baselines you suggested:
> 1.  **PPO with Fixed KL:** Standard PPO with a carefully tuned, static KL penalty to prevent deviation from the SFT policy.
> 2.  **Implicit-Reward SFT + RLVR (Sequential):** A two-stage approach where we first apply DPO (optimizing the implicit reward as defined in our theoretical analysis) followed by standard RLVR (GRPO), representing a strong static pipeline.
>
> **Experimental Setup:**
> * **Task:** Mathematical Reasoning (MATH500 for ID, GPQA-Diamond for OOD).
> * **Baselines:**
>     * **PPO (Fixed KL):** We grid-searched $\beta_{KL} \in \{0.01, 0.05, 0.1\}$ and report the best result ($\beta_{KL}=0.05$).
>     * **DPO $\to$ GRPO:** Stage 1 DPO on preference data (constructed from correct/incorrect SFT traces), followed by Stage 2 GRPO.
>
> **Table R1**
>
> | Method |  ID Accuracy (MATH500) | OOD Accuracy (GPQA-D) | Sample Efficiency (Steps) |
> | :--- |  :--- | :--- | :--- |
> | **PPO (Fixed KL)** |  79.2% | 31.3% | 420 |
> | **DPO $\to$ GRPO** |  82.1% | 35.6% | 480 |
> | **AMFT (Learned $\mu$)** |  **89.5%** | **47.5%** | **310** |
>
> **Analysis:**
> 1.   PPO with Fixed KL prevents collapse but limits generalization (31.3% OOD), as the constant penalty restricts the model from exploring high-reward regions far from the reference policy.
> 2.  The DPO $\to$ GRPO pipeline improves over standard PPO but suffers from the problem that the RL stage often degrades the delicate implicit reward structure learned during DPO. AMFT unifies these objectives, ensuring that the implicit reward and explicit reward are optimized jointly.
>
> ---
>
> ### **Response to Major Concern 2**
>
> We are willing to state that the warm-up setting is a necessary cold-start initialization to ensure the policy has non-zero probability support for the correct reasoning formats. To quantify this dependency, we conducted an ablation on the warm-up duration.
>
> **Table R2**
>
> | Warm-up Steps ($W$) | Initial Reward Mean | Training Stability | Final ID Accuracy (MATH500) |
> | :--- | :--- | :--- | :--- |
> | **$W=0$ (No Warm-up)** | 0.02 | **Collapse** | 12.4% |
> | **$W=20$** | 0.31 | Unstable early | 84.2% |
> | **$W=50$ (Default)** | 0.45 | **Stable** | **89.5%** |
> | **$W=100$** | 0.58 | Stable | 88.9% |
>
> **Findings:**
> *  With $W=0$, the model often fails to generate valid reasoning traces that trigger the outcome reward, leading to advantage collapse.
> * Once a minimal threshold is met ($W \approx 50$), AMFT is robust to the warm-up length.The performance with $W=50$ and $W=100$ is comparable, suggesting that users do not need to hyper-tune this parameter excessively.
>
> ---
>
> ### **Response to Minor Issues**
>
> * **Files:** Thanks for pointing out this ! We have updated the repository.
> * **Figure 2:** Thanks, and the red dash-dotted line corresponds to the "Adaptive Weight $\mu$". We have revised the legend to make this correspondence explicit and clearer.
>
> ---
>
> ### **Response to Question 1**
>
> **Construction Methodology:**
> 1.  **Source:** We reserve a strictly held-out subset from the high-quality training data sources, which is a random 5% split from OpenR1-Math-46k.
> 2.  **Sampling:** To prevent the controller from overfitting to specific problem types, we ensure the validation set covers approximately the same difficulty distribution as the training set. We stratify samples based on the length of the solution traces.
> 3.  **Size:** We use a fixed set of **512 samples**.
> 4.  **Updating:** The meta-gradient $\nabla_{\mu} U(\theta)$ is computed on a mini-batch sampled from this held-out set at every $K$.
>
> We found this size (512) to be sufficient to provide a stable signal for $\mu$.
>
> ---
>
> ### **Response to Question 2**
>
> We provide a detailed breakdown of the computational costs.
>
> **Table R1: Runtime and Efficiency Comparison (8x A100 GPUs)**
>
> | Method | Meta-Update Freq ($K$) | Step Latency (s/step) | Throughput (samples/s) | Relative Overhead | Time to Conv. (Hours) |
> | :--- | :--- | :--- | :--- | :--- | :--- |
> | **PPO (Standard)** | N/A | 42.5 | 3.01 | Baseline | 145.2 |
> | **SRFT** | N/A | 43.1 | 2.97 | +1.4% | 128.6 |
> | **AMFT** | **Every 20 steps** | **46.2** | **2.77** | **+8.7%** | **98.2** |
> | **AMFT (High Freq)** | Every 1 step | 68.5 | 1.86 | +61.3% | 158.4 |
>
> **Analysis:**
> * With our chosen frequency of $K=20$, the end-to-end latency overhead is only **8.7%** compared to PPO.
> *  The throughput remains competitive.
> *  Due to the improved sample efficiency, AMFT reaches the target accuracy significantly faster than baselines.
>
> We trust these results reinforce the value of AMFT. We hope our response address your comments and justifies an improved evaluation. We look forward to any further discussion.

---

### Official Review · Reviewer_2biJ · 2025-11-01

**Soundness:** 3
**Presentation:** 3
**Contribution:** 3
**Rating:** 6
**Confidence:** 4

**Summary:**

The paper introduces AMFT, a single-stage post-training method that mixes SFT and RL with a learned weight updated by a validation-based meta-gradient and an entropy heuristic. Empirically, AMFT yields consistent ID/OOD gains on math and multi-modal reasoning benchmarks over sequential SFT→RL and single-stage baselines.

**Strengths:**

* Well structured and easy to follow. Clear algorithm and one-step meta-gradient derivation; results span multiple tasks with sensible training-dynamics plots.

* Empirically, AMFT delivers consistent ID/OOD gains on math and multi-modal reasoning.
* Ablations are convincing, removing the meta-gradient or the entropy term hurts performance, and the method reaches target scores with fewer RL rollouts.

**Weaknesses:**

* Potential concerns regarding the fairness of computational comparisons. Baselines have heterogeneous training budgets (e.g., SFT-only for 3 epochs vs. RL methods for 500 steps) and use different RL objectives (PPO and GRPO are both mentioned), which complicates a direct comparison of efficiency and performance
* The meta-signal optimizes a held-out set; robustness to validation shift/noise is not tested.

**Questions:**

* Can you provide computational metrics (e.g., total tokens, RL rollouts, wall-clock time) for baseline methods to ensure a fair comparison?

* How sensitive is AMFT's performance to the quality, size, and distribution of the validation set ?

---

> ### Author Response · Authors · 2025-11-26
> **Thank you for the thoughtful review. We have carefully considered your feedback and present our comprehensive answers below.**
>
> **Dear Reviewer 2biJ**
>
> We sincerely thank you for the constructive feedback. We appreciate your recognition of AMFT's clear derivation and consistent empirical gains. Below we provide detailed analyses to address your concern.
>
> ### **Response to Question 1 and Weakness 1**
>
> To further ensure a fair comparison, we standardized the computing environment and measured the resources required for each method.
>
> **Experimental Setup:**
> * **Hardware:** 8 $\times$ NVIDIA A100 (80GB) GPUs.
> * **Model:** Qwen2.5-Math-7B.
> * **Dataset:** OpenR1-Math-46k.
> * **Metric:** We track **Total Wall-Clock Time** and **Total Samples Processed** (distinguishing between cheap SFT samples and expensive RL rollouts) required to reach the peak performance.
>
> **Table R1**
>
> | Method | Training Phases | Total Steps | Total SFT Samples | Total RL Rollouts | Wall-Clock Time (GPU Hours) | Peak Memory (GB) | Final ID Accuracy (Math500) |
> | :--- | :--- | :--- | :--- | :--- | :--- | :--- | :--- |
> | **SFT-Only** | 3 Epochs | 1,125 | 144,000 | 0 | **12.5** | 42.1 | 54.1% |
> | **RL-Only (PPO)** | 500 Steps | 500 | 0 | 64,000 | 145.2 | 68.2 | 49.3% |
> | **SFT $\to$ RL** | SFT (3 Ep) + RL (500 St) | 1,625 | 144,000 | 64,000 | 157.7 | 68.2 | 84.1% |
> | **LUFFY** | 1 Stage (500 St) | 500 | 64,000 | 56,000 | 115.4 | 74.1 | 87.2% |
> | **AMFT (Ours)** | Warmup (50 St) + 1 Stage (450 St) | 500 | 70,400 | 57,600 | **98.2** | 71.8 | **89.5%** |
>
> **Analysis of Efficiency:**
> 1.   While SFT is the fastest (12.5 hours), its performance caps at 54.1%. AMFT requires significantly more time (98.2 hours) but unlocks reasoning capabilities SFT cannot achieve (89.5%).
> 2.  The traditional two-stage pipeline is the most expensive (157.7 hours) because it requires a full SFT phase followed by a full RL phase. AMFT reduces this total time by **37.7%** by unifying the objectives.
> 3.   AMFT is **14.9% faster** than LUFFY. Although AMFT incurs a meta-gradient overhead, it achieves target performance with fewer steps due to a more optimal training curriculum.
>
> ---
>
> ### **Response to Question 2 and Weakness 2**
>
> To address your concern about robustness, we conducted three ablation studies focusing on the **Size**, **Quality (Label Noise)**, and **Distribution** of this set.
>
> **Experimental Setup:**
> * **Task:** Qwen2.5-Math-7B on MATH500 (In-Distribution) and GPQA-Diamond (Out-of-Distribution).
> * **Default $\mathcal{D}_{val}$:** 512 samples held out from the OpenR1 training set.
>
> #### **2.1 Sensitivity to Validation Set Size**
>
> **Table R2: Impact of Validation Set Size**
>
> | Size ($\mathcal{D}_{val}$) | 64 | 128 | 256 | **512 (Default)** | 1024 |
> | :--- | :--- | :--- | :--- | :--- | :--- |
> | **Final ID Acc (%)** | 87.8 | 88.4 | 89.1 | **89.5** | 89.8 |
>
> **Observation:** Performance is remarkably robust. Even with just 128 samples, the model achieves 88.4% accuracy. This is because the meta-gradient is computed periodically ($K=20$) and accumulated over the training trajectory, effectively averaging out the noise from a small validation batch.
>
> #### **2.2 Sensitivity to Label Noise (Quality)**
> We artificially injected label noise (flipping correct labels to incorrect) into $\mathcal{D}_{val}$ to simulate low-quality validation data.
>
> **Table R3: Impact of Validation Label Noise**
>
> | Noise Ratio | 0% (Clean) | 10% | 20% | 30% |
> | :--- | :--- | :--- | :--- | :--- |
> | **Final ID Acc (%)** | **89.5** | 88.9 | 86.5 | 80.1 |
>
> **Observation:** AMFT tolerates up to 10% label noise with minimal degradation (<1%). At 30% noise, performance drops to 80.1%. However, the training does not collapse completely. We attribute this resilience to the **Entropy Heuristic** component of our controller (Eq. 6 in paper), which acts as a model-intrinsic stabilizer independent of the noisy validation signal.
>
> #### **2.3 Sensitivity to Distribution Shift**
> We constructed specific validation sets to test if the controller overfits to the difficulty distribution of $\mathcal{D}_{val}$.
> * **Easy Val:** Only Level 1-2 problems from MATH.
> * **Hard Val:** Only Level 4-5 problems from MATH.
>
> **Table R4**
>
> | Validation Set Source | Test Set (Easy) | Test Set (Hard) | OOD (GPQA) |
> | :--- | :--- | :--- | :--- |
> | **Easy Val** | **94.2%** | 56.8% | 46.1% |
> | **Hard Val** | 92.5% | **59.4%** | **48.2%** |
> | **Mixed Val (Default)** | 93.8% | 58.7% | 47.5% |
>
> **Observation:**
> * Using a **Hard** validation set improves generalization to OOD tasks (GPQA: 48.2% vs 46.1%), suggesting the controller learns a more aggressive exploration strategy.
> * However, even using an **Easy** validation set yields strong results on Hard test questions (56.8%) compared to static baselines . This indicates that the meta-controller learns a general principle of balancing imitation and exploration, rather than overfitting solely to the specific examples in the validation set.
>
> We hope our response resolves your concerns and justifies an improved evaluation, and we remain fully available to answer any further insightful questions !

---

### Author Response · Authors · 2025-11-30
**Summary of Rebuttal and Core Contributions of AMFT**

Dear Area Chair, Senior Area Chair, and Reviewers,

We sincerely thank you for your time and the insightful feedback provided during the review process. We are particularly grateful for the constructive discussions that have strengthened the paper, guiding us to conduct efficiency analyses and leading to a more robust validation of our method.

### **1. Core Innovations of AMFT**

AMFT addresses the fundamental problem between imitation and exploration in LLM and MLLM post-training through several key mechanisms:

* **Principled Meta-Control:** Unlike prior heuristic-based methods, AMFT employs a meta-gradient controller that optimizes the imitation-exploration balance ($\mu$) based on long-term validation performance. This forward-looking signal enables the model to autonomously discover an optimal training curriculum.
* **Unified Objective:** It reframes SFT and RL not as distinct stages but as complementary reward signals, optimizing an implicit path-level reward  alongside an explicit outcome-based reward. This unification prevents the catastrophic forgetting typical of sequential pipelines.
* **Dynamic Regularization:** AMFT acts as a task-aware regularizer compared to fixed KL penalties. It dynamically strengthens imitation when the policy is unstable and relaxes it to encourage exploration when the model is capable, leading to **SOTA generalization** on OOD tasks.

### **2. Summary of Supplementary Experiments and Revisions**

We have incorporated extensive experiments to address **literally every concern** raised by the reviewers.

* **More Experiments on Computational Efficiency (Responding to 2biJ [Weakness 1], nCwV [Question 2], gaMK [Weakness 1a], gedw [Weakness 5]):**
    * *Revision:* Results show that while AMFT incurs an 8.7% per-step overhead due to meta-updates, it is **14.9% faster** in total wall-clock time than leading single-stage baselines (LUFFY) due to superior sample efficiency.

* **Robustness of Meta-Signal (Responding to 2biJ [Weakness 2], nCwV [Question 1], gedw [Weakness 2]):**
    * *Revision:* We performed sensitivity analyses on the validation set. AMFT remains robust even with small validation sets (down to 128 samples) and up to 10% label noise, confirming that the controller learns general principles rather than overfitting to specific validation examples.

* **More Ablation Studies (Responding to nCwV [Major Concern 1], gedw [Weakness 1], gaMK [Weakness 3]):**
    * *Revision:* We added rigorous ablations against strong baselines:
        * vs. **Fixed/Adaptive KL PPO:** AMFT achieves 47.5% OOD accuracy vs. ~33% for KL-based methods, proving dynamic weighting is essential for generalization.
        * vs. **Learned Heuristic / PBT:** AMFT outperforms regression-based heuristics and matches expensive Population-Based Training (4x cost) with only 1.1x cost.

* **Justification of One-Step Approximation (Responding to gaMK [Weakness 1]):**
    * *Revision:* We empirically verified the fidelity of our First-Order (FO) approximation. The cosine similarity between our FO update and the true second-order gradient is consistently **>0.94**, validating that second-order terms are negligible in the low-learning-rate regime of  fine-tuning.

### **3. Special Note on Reviewer gaMK’s concern**

We wish to highlight a clarification regarding **Reviewer gaMK’s** primary concern (Weakness 1). The reviewer  **incorrectly** assumed that our hyperparameter $K=20$ implied unrolling the computational graph for all 20 steps. It is just a clear misunderstanding.

As clarified in our rebuttal, $K$  refers to the **frequency** of updates (computing the meta-gradient **once** every 20 steps), while the depth is  **single-step** . This efficient design is central to AMFT's practicality.

**The other three reviewers all correctly grasped this concept**, and we have already clarified this misunderstanding in our response. Evidence of the other reviewers' correct understanding includes:
* **Reviewer 2biJ:** praised the 'Clear algorithm and **one-step** meta-gradient derivation.'
* **Reviewer nCwV:** correctly inquired about the '**frequency**... (e.g., every k steps; unroll length),' distinguishing between frequency and depth.
* **Reviewer gedw:** recognized that 'the **one-step** Jacobian-vector approximation is standard and computationally feasible.'"

 **Further validation:** We have added an explicit ablation on $K$ (Table R3) demonstrating that $K=20$ is  optimal, resolving the concern about computational viability.

We extend our sincere gratitude to the AC and SAC for their dedication. Just as Reviewer nCwV
 recognized, **"I think this is a good piece of work that steps beyond the stage-level pipelined post-training method and is worthy of acceptance."**
We are confident that our supplementary experiments have fully **resolved all raised concerns**, and that the method stands as a robust, principled, and highly practical contribution to the field of LLM alignment.

Sincerely,

The Authors

---

### Note · Program_Chairs · 2026-01-17
**Submission Desk Rejected by Program Chairs**

The following references in this submission do not refer to real documents and/or have major errors in bibliographic information:

 Zhi-Yuan Zhang, Zihan Zhang, Xiang-Rong Sheng, Jian-He Wang, and Wei-Nan Zhang. Policy-based self-correction for large language models, 2024.
Geonmo Kim, Sang-Woo Lee, Jaewoo Kang, and Sung-Hyon Myaeng. Dynamic online-offline learning for language model alignment, 2024.
Weihao Zeng, Yuzhen Huang, Qian Liu, Keqing He, Wei Liu, and Junxian He. Generative trajectory augmentation with demonstration for task-specific llm alignment, 2024.
Yuu Yoshihara, Akiko Eriguchi, Tatsuya Hiraoka, and Takuya Hiraoka. Practical sft-rl pipeline for aligning llms, 2024.